# Mapping a 50-spin-qubit network through correlated sensing

G. L. van de Stolpe [1,2], D. P. Kwiatkowski [1,2], C. E. Bradley [1,2], J. Randall[1,2], M. H. Abobeih[1,2], S. A. Breitweiser[3], L. C. Bassett [3], M. Markham[4], D. J. Twitchen[4] & T. H. Taminiau [1,2] ✉

Spins associated to optically accessible solid-state defects have emerged as a versatile platform for exploring quantum simulation, quantum sensing and quantum communication. Pioneering experiments have shown the sensing, imaging, and control of multiple nuclear spins surrounding a single electron spin defect. However, the accessible size of these spin networks has been constrained by the spectral resolution of current methods. Here, we map a network of 50 coupled spins through high-resolution correlated sensing schemes, using a single nitrogen-vacancy center in diamond. We develop concatenated double-resonance sequences that identify spin-chains through the network. These chains reveal the characteristic spin frequencies and their interconnections with high spectral resolution, and can be fused together to map out the network. Our results provide new opportunities for quantum simulations by increasing the number of available spin qubits. Additionally, our methods might find applications in nano-scale imaging of complex spin systems external to the host crystal.

Optically interfaced spin qubits associated to defects in solids provide a versatile platform for quantum simulation[1], quantum networks[2,3] and quantum sensing[4-6]. Various systems are being explored[7], including defects in diamond[1-3,8,9], silicon carbide[10,11], silicon[12,13], hexagonal boron nitride (hBN)[14], and rare-earth ions[15]. The defect electron spin provides a qubit with high-fidelity control, optical initialization and readout, and a (long-range) photonic quantum network interface[2,3]. Additionally, the electron spin can be used to sense and control multiple nuclear spins surrounding the defect[15-17]. This additional network of coupled spins provides a qubit register for quantum information processing, as well as a test bed for nano-scale magnetic resonance imaging[18-23]. Examples of emerging applications are quantum simulations of many-body physics[1,24-27], as well as quantum networks[2,3], where the nuclear spins provide qubits for quantum memory[28], entanglement distillation[29], and error correction[30-32].

State-of-the-art experiments have demonstrated the imaging of spin networks containing up to 27 nuclear spins[18,19,33-35]. The ability to map larger spin networks can be a precursor for quantum simulations that are currently intractable, would provide a precise understanding of the noise environment of spin-qubit registers[32,36,37], and might contribute towards efforts to image complex spin systems outside of the host material[20-23,38,39]. A key open challenge for mapping larger networks is spectral crowding, which causes overlapping signals and introduces ambiguity in the assignment of signals to individual spins and the interactions between them.

Here, we develop correlated sensing sequences that measure both the network connectivity as well as the characteristic spin frequencies with high spectral resolution. We apply these sequences to map a 50-nuclear-spin network comprised of 1225 spin-spin interactions in the vicinity of a nitrogen-vacancy (NV) center in diamond. The

[1]QuTech, Delft University of Technology, PO Box 5046, 2600 GA Delft, The Netherlands. [2]Kavli Institute of Nanoscience Delft, Delft University of Technology, PO Box 5046, 2600 GA Delft, The Netherlands. [3]Quantum Engineering Laboratory, Department of Electrical and Systems Engineering, University of Pennsylvania, 200 South 33rd Street, Philadelphia, PA 19104, USA. [4]Element Six Innovation, Fermi Avenue, Harwell Oxford, Didcot, Oxfordshire OX11 0QR, UK. ✉e-mail: t.h.taminiau@tudelft.nl

key concept of our method is to concatenate double-resonance sequences to measure chains of coupled spins through the network. The mapping of spin chains removes ambiguity about how the spins are connected and enables the sensing of spins that are farther away from the electron spin-sensor in spectrally crowded regions. These results significantly increase the size and complexity of the accessible spin network. Additionally, our methods are applicable to a wide variety of systems and might inspire future methods to magnetically image complex samples such as individual molecules or proteins[22,33,39].

## Results

### Spin-network mapping

We consider a network of $N$-coupled nuclear spins in the vicinity of a single electron spin that acts as a quantum sensor[18,19]. The effective dynamics of the nuclear-spin network, with an external magnetic field along the z-axis, are described by the Hamiltonian (see Supplementary Note 1):

$$\hat{H} = \sum_{i=1}^{N} A_i \hat{I}_z^{(i)} + \sum_{i=1}^{N} \sum_{j=i+1}^{N} C_{ij} \hat{I}_z^{(i)} \hat{I}_z^{(j)}, \qquad (1)$$

where $\hat{I}_z^{(i)}$ denotes the nuclear Pauli spin-$\frac{1}{2}$ operator for spin $i$, $A_i$ are the precession frequencies associated with each spin, and $C_{ij}$ denotes the nuclear-nuclear coupling between spin $i$ and $j$. The frequencies $A_i$ might differ due to differences in species (gyromagnetic ratio), the local magnetic field and spin environment, and due to coupling to the sensor electron spin. Our goal is to extract the characteristic spin frequencies $A_i$ and spin-spin couplings $C_{ij}$ that capture the structure of the network.

Figure 1 shows an example network, with colored disks denoting frequency regions, and numbered dots inside signifying spins at these frequencies. Although in principle all spins are coupled to all spins, we draw edges only for strong, resolvable, spin-spin couplings, defined by: $C_{ij} \gtrsim 1/T_2$, where $T_2$ is the nuclear Hahn-echo coherence time (~0.5 s)[16]. The network connectivity constitutes the presence (or absence) of such resolvable couplings. In general, the number of frequency disks is smaller than the number of spins, as multiple spins might occupy the same frequency region (i.e., overlap in frequency).

State-of-the-art spin network mapping relies on isolating individual nuclear-nuclear interactions through spin-echo double resonance (SEDOR)[18]. Applying simultaneous echo pulses at frequencies $A_i$ and $A_j$ preserves the interaction $C_{ij}$ between spins at $A_i$ and $A_j$, while decoupling them from (quasi-static) environmental noise and the rest of the network so that the coupling $C_{ij}$ is encoded in the nuclear-spin polarization with high spectral resolution (set by the nuclear $T_2$-time rather than $T_2^\star$-time). The signal is acquired by mapping the resulting nuclear-spin polarization, for example at frequency $A_i$, on the NV electron spin and reading it out optically[16]. Such a measurement yields a correlated list of three frequencies {$A_i$, $C_{ij}$, $A_j$} (Fig. 1a). If all spins are spectrally isolated so that the $A_i$ do not overlap, these pairwise measurements completely characterize the network.

However, due to their finite spectral line widths (set by $1/T_2^\star$), multiple spin frequencies $A_i$ may overlap (indicated by multiple spins occupying a disk). This introduces ambiguity when assigning measured couplings to specific spins in the network, and causes complex overlapping signals, which are difficult to resolve and interpret[18,19]. Figure 1b shows an example where pairwise measurements break down; spins 2 and 5 overlap in frequency ($A_2 \approx A_5$). Applying pairwise SEDOR between frequencies $A_1$, $A_3$, $A_4$ and a frequency that overlaps with $A_2$ and $A_5$ returns three independent pairwise correlations: {$A_1$, $C_{12}$, $A_2$}, {$A_3$, $C_{23}$, $A_2$} and {$A_4$, $C_{45}$, $A_5$}. Crucially, however, such measurements cannot distinguish this uncoupled 2-spin and 3-spin chain (Fig. 1b) from a single 4-spin network (with a single central spin at $A_2$), nor from a network of 3 uncoupled 2-spin chains (three spectrally overlapping spins). Without introducing additional a-priori knowledge or assumptions about the system, pairwise measurements cannot be assigned to specific spins and are thus insufficient to reconstruct the network[18].

Our approach is to measure connected chains through the network and combine these with high-resolution spin-frequency measurements. First, spin-chain sensing (detailed in the "Spin-chain sensing" section) correlates multiple frequencies and spin-spin couplings, directly accessing the underlying network connectivity, and thus reducing ambiguity due to (potential) spectral overlap. Consider the previous example: by probing the correlation between the three frequencies $A_1$, $A_2$, and $A_3$ in a single measurement, we directly reveal that Spin 1 and Spin 3 are connected to the same spin at $A_2$ (Spin 2).

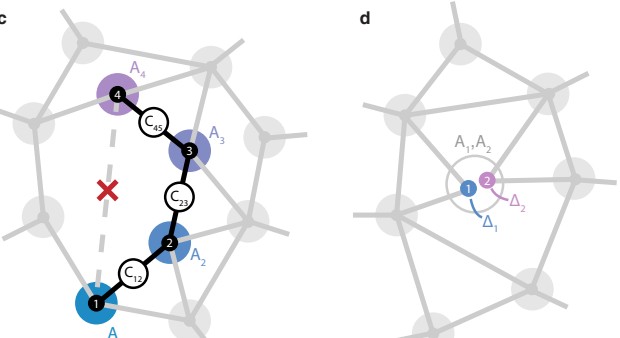

**Fig. 1 | Mapping spin networks.** Graph representing a spin network, where vertices denote spins and edges denote spin-spin interactions ($C_{ij}$). Spins are distributed among spectral regions (colored disks) by their precession frequency ($A_i$). **a** If all spin frequencies are unique (one spin in each disk), the network can be mapped by measuring only pairwise interactions ($C_{12}$) between frequencies ($A_1$, $A_2$). **b** If spins spectrally overlap (e.g., spins 2 and 5 with $A_2 \approx A_5$) due to the finite linewidth set by the dephasing time $T_2^\star$, pairwise measurements alone are ambiguous when assigning interactions to specific spins. By measuring chains (e.g., through $A_1$, $A_2$, $A_3$) we directly retrieve the connectivity of the network. **c** We also exploit spin chains to measure interactions between spins that are otherwise challenging to access. As an example, couplings belonging to Spin 4 are not directly accessible

from the spin at $A_1$ - due to spectral crowding or negligible couplings - but can be obtained through a chain. **d** Finally, we complement the spin chains with a correlated double-resonance method that enhances the spectral resolution for the spin-frequency shifts ($\Delta_i$) from ~$1/T_2^\star$ to ~$1/T_2$, so that spectrally overlapping spins can also be resolved directly. This figure shows a conceptual network with vertices organized in frequency space. In Supplementary Fig. 1, we discuss the specific relations between frequency and spatial position for the experimental system considered here: an NV center in diamond and surrounding $^{13}$C-spin network, for which increased spectral crowing (panels **b**–**d**) naturally occurs for $^{13}$C spins that are farther away from the NV center.

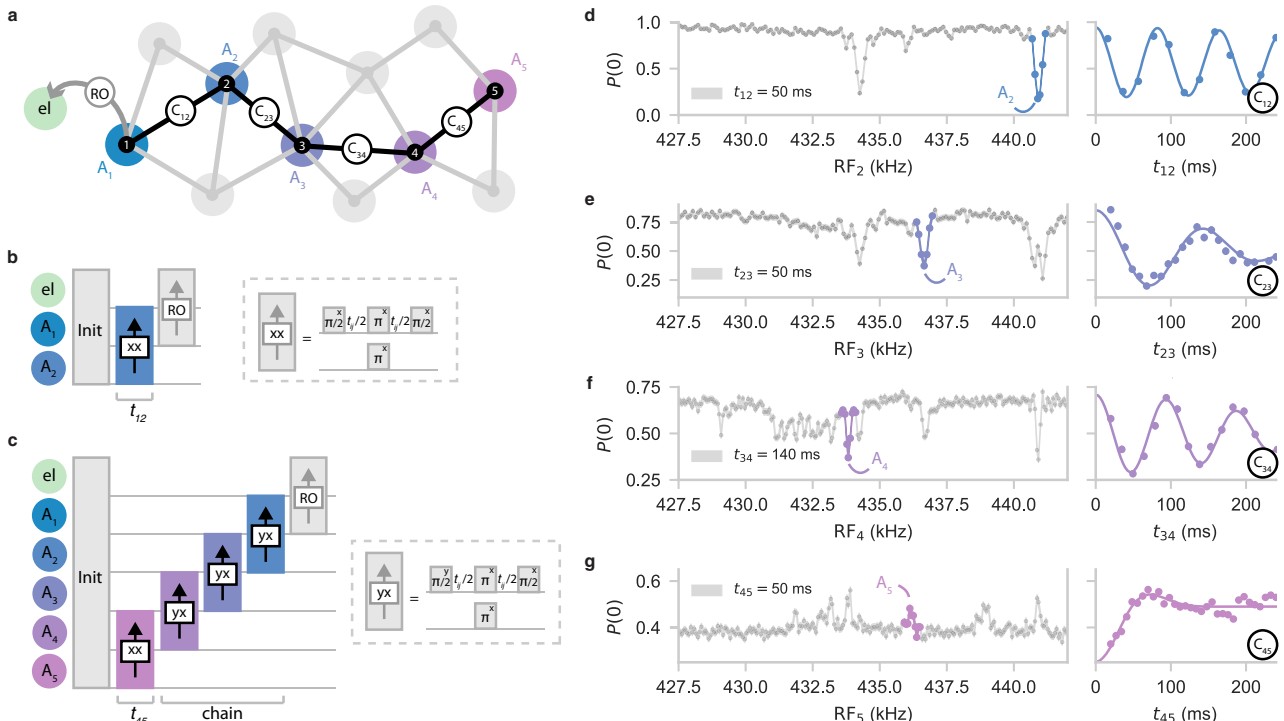

**Fig. 2 | Sensing spin chains. a** Schematic of an $N = 5$ nuclear-spin chain through different spectral regions $\{A_1, A_2, A_3, A_4, A_5\}$ (colored disks), starting from the NV electron spin ('el'). Even though there might be multiple spins at each of the nuclear frequencies, only a single one is connected to this chain. **b** Pulse sequence (see the "Methods" section) for the prototypical $N = 2$ sequence (SEDOR)[18]. **c** Pulse sequence for sensing a chain of $N = 5$ nuclear spins, correlating 5 spin frequencies and 4-spin-spin interactions. In this case, the RF frequency (RF$_5$) and free evolution time ($t_{45}$) are varied to probe the connections of the spin at $A_4$ to other spins. The resulting signal is mapped back via concatenated SEDOR sequences and finally read out ('RO') through the electron spin (see "Methods" section). **d–g** Experimental data, sweeping the frequency RF$_N$ of the recoupling pulse (left) to detect the frequencies of spins coupled to Spin 1, and varying the free evolution time $t_{N-1,N}$ (right) to extract their coupling strengths (for $N = \{2, 3, 4, 5\}$). For the frequency sweeps, evolution times $t_{ij}$ are selected a-priori (annotated). Colored highlights denote the signals due to the spins in the chain and solid lines are fits to the data (see Supplementary Note 3). The signal in the bottom panel is inverted, due to the coupling $C_{34}$ being negative.

Such a spin-chain measurement yields a correlated list of 5 frequencies: $\{A_1, C_{12}, A_2, C_{23}, A_3\}$, characterizing the 3-spin chain. Applying the same method but now with spin 4 ($A_3 \leftarrow A_4$) reveals that it is not connected to Spin 2, but couples to another spin (spin 5) that overlaps in frequency with Spin 2.

Second, spin-chain sensing enables measuring couplings that are otherwise challenging to access, enabling exploration further into the network. Consider the case where starting from some spin (e.g. Spin 1 in Fig. 1c) it is challenging to probe a part of the network, either because the couplings to Spin 1 are too weak to be observed or spectral crowding causes signals to overlap. The desired interactions (e.g. those belonging to Spin 4 in Fig. 1c) can be reached by constructing a spin chain, in which each link is formed by a strong and resolvable spin-spin interaction. The chain iteratively unlocks new spins that can be used as sensors of their own local spatial environment.

Finally, we combine the spin-chain measurements with a correlated double-echo spectroscopy scheme that increases the resolution with which different $A_i$ are distinguished from $\sim 1/T_2^\star$ to $\sim 1/T_2$ (Fig. 1d). This directly reduces spectral overlap of spin frequencies, further removing ambiguity.

In principle, the entire network can be mapped by expanding and looping a single chain. In practice, measuring limited-size chains is sufficient. An $N$-spin-chain measurement yields a correlated list of $N$ spin frequencies $A$, alongside $N-1$ coupling frequencies $C$, which quickly becomes uniquely identifiable, even when some spin frequencies in the network are degenerate. This allows for the merging of chains that share a common section to reconstruct the network (see "Methods" section).

## Experimental system

We demonstrate these methods on a network of 50 $^{13}$C spins surrounding a single NV center in diamond at 4 K. The NV electron spin is initialized and measured optically and is used as the sensor spin[18]. We employ dynamical decoupling sequences to sense nuclear spins at selected frequency bands, using sequences with and without radio-frequency driving (DDRF) of the nuclear spins to ensure sensitivity in all directions from the NV (see "Methods" section)[16,18]. The nuclear spins are polarized via the electron spin, using global dynamical-nuclear-polarization techniques (PulsePol sequence[1,40]), or by selective projective measurements or SWAP gates[16,18].

The $^{13}$C nuclear-spin frequencies are given by $A_i = \omega_L + m_s \Delta_i$, with $\omega_L$ the global Larmor frequency and $\Delta_i$ a local shift due to the hyperfine interaction with the NV center (see for example ref. 41 and Supplementary Note 1). Here, we neglected corrections due to the anisotropy of the hyperfine interaction, which are treated in Supplementary Note 4. The experiments are performed with the electronic spin in the $m_s = \pm 1$ states. Because, for the spins considered, $\Delta_i$ is typically two to three orders of magnitude larger than the nuclear-nuclear couplings $C_{ij}$, nuclear-spin flip-flop interactions are largely frozen, and Eq. (1) applies (Supplementary Note 1).

In the NV-nuclear system, spectral crowding forms a natural challenge for determining the spin network structure. The spin frequencies are broadened by the inhomogeneous linewidth $\sim 1/T_2^\star$, which is mainly set by the coupling to all other nuclear spins. A limited number of nuclear spins close to the NV center are spectrally isolated (defined as: $|A_i - A_j| > 1/T_2^\star \; \forall \, j$), making them directly accessible with electron-nuclear gates[16,18], and making pairwise measurements sufficient to map the interactions. However, the hyperfine interaction, and

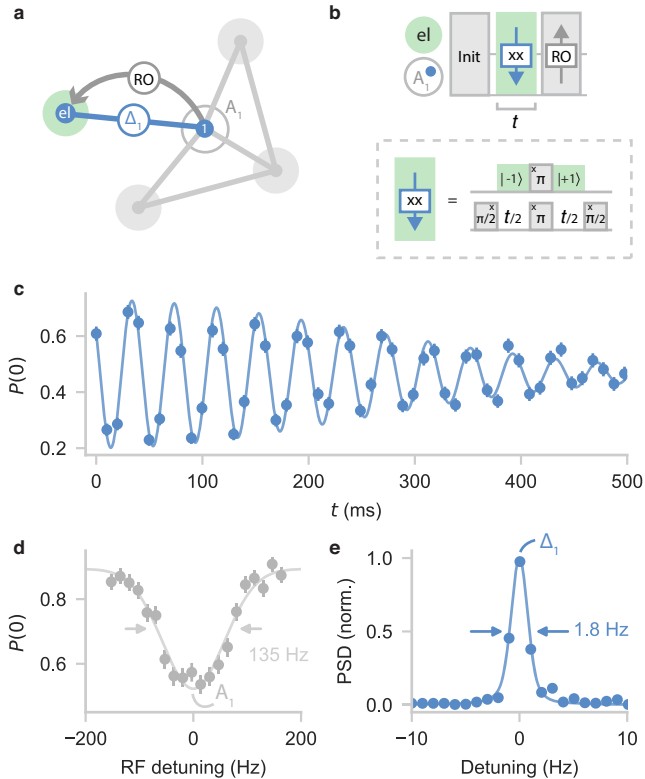

**Fig. 3 | Electron-nuclear double resonance. a** The nuclear-spin frequencies $A_i$ are shifted by the hyperfine interaction with the electron spin $\Delta_i$ (blue dotted line). Performing double resonance between the nuclear spin and the electron spin (mint green) retains this interaction while decoupling from quasi-static noise. **b** Pulse sequence for measuring $\Delta_1$ for a spin at $\approx A_1$. The nuclear spin undergoes a double-resonance sequence, picking up a phase (downward arrow) from the interaction with the electron spin, whose population is synchronously transferred from the $|-1\rangle$ to the $|+1\rangle$ state. Finally, the signal is read out (denoted 'RO') via the electron spin (see "Methods" section) **c** Time-domain signal of $\Delta_1 = 14549.91(5)$ Hz (under-sampled), with a coherence time of $T_2 = 0.36(2)$ s, fitted by a sinusoid with Gaussian decay. **d** Zoom-in of spectroscopy data as in Fig. 2d, showing a broad resonance (135 Hz FWHM), limited by the nuclear $T_2^\star$-time. **e** Power spectral density (PSD) of (**c**), showing a linewidth that is ~75 times improved compared to (**d**).

thus $\Delta_i$, decreases with distance ($\sim r^{-3}$), resulting in an increasing spectral density for lower $\Delta_i$ (larger distance). Interestingly, there exists a spectrally crowded region ($|A_i - A_j| < 1/T_2^\star$) for which nuclear spins still do not couple strongly to other spins in the same spectral region ($C_{ij} \lesssim 1/T_2 \forall j$), for example when they are on opposite sides of the NV center. Contrary to previous work[18], the methods outlined in the "Spin-network mapping" section allow us to measure interactions between spins in the spectrally crowded region (see Supplementary Note 2), unlocking a part of the network that was previously not accessible.

## Spin-chain sensing
We experimentally demonstrate the correlated sensing of spin chains up to five nuclear spins (Fig. 2), by sweeping a multi-dimensional parameter space (set by 5 spin frequencies and 4 spin-spin couplings). We start by polarizing the spin network[1,40] and use the electron spin to sense a nuclear spin (Spin 1) at frequency $A_1$, which marks the start of the chain.

First, we perform a double-resonance sensing sequence (Fig. 2b) consisting of a spin-echo sequence at frequency $A_1$ and an additional $\pi$-pulse at frequency $RF_2$. The free evolution time $t_{12}$ is set to 50 ms, to optimize sensitivity to nuclear-nuclear couplings (typically ~10 Hz). By sweeping $RF_2$, strong connections ($C_{1j} \gg 1/T_2$) are revealed through

dips in the coherence signal of Spin 1 (Fig. 2d, left). We select a connection to a spin at $RF_2 = A_2$ (Spin 2) and determine ($C_{12}$) by sweeping $t_{12}$ (Fig. 2d, right).

Next, we extend the chain. To map the state of Spin 2 back to the electron sensor through Spin 1, we change the phase of the first $\frac{\pi}{2}$-pulse (labeled 'yx') and set $t_{12} = 1/(2C_{12})$ to maximize signal transfer (see Supplementary Note 3). We then insert a double-resonance block for frequencies $RF_2 = A_2$ and $RF_3$ in front of the sequence (Fig. 2c, e, left) to explore the couplings of Spin 2 to the network. This concatenating procedure can be continued to extend the chain, with up to 5 nuclear spins shown in Fig. 2. In general, the signal strength decreases with increasing chain length, as it is set by a combination of the degree of polarization and decoherence ($T_2$ relative to $C_{ij}$) of all spins in the chain (See Supplementary Note 3). This limits the chain lengths that can be effectively used.

By mapping back the signal through the spin chain, the five spin frequencies and the 4 coupling frequencies are directly correlated: they are found to originate from the same branch of the network. As spins are now characterized by their connection to the chains, rather than by their individual, generally degenerate, frequencies (Fig. 1b), they can be uniquely identified. Additionally, the chains enable measuring individual spin-spin couplings in spectrally crowded regions (Fig. 1c). As an example, the expected density of spins at frequency $A_4$ is around 30 spins per kHz (Supplementary Fig. 6), making Spin 4 challenging to access directly from the electron spin. However, because Spin 3 probes only a small part of space, Spin 4 can be accessed through the chain, as demonstrated by the single-frequency oscillation in Fig. 2f. Another advantage over previous methods[18] is that our sequences are sensitive to both the magnitude and the sign of the couplings, at the cost of requiring observable polarization of the spins in the chain. The sign of the couplings provides additional information for reconstructing the network (Fig. 2g).

## High-resolution measurement of spin frequencies
While the sensing of spin-chains unlocks new parts of the network and reduces ambiguity by directly mapping the network connections, the spectral resolution for the spin frequencies ($A_i$) remains limited by the nuclear inhomogeneous dephasing time $T_2^\star \sim 5$ ms[16]. Next, we demonstrate high-resolution ($T_2$-limited) measurements of the characteristic spin-frequency shifts $\Delta_i$. These frequencies provide a way to label spins, and thus further reduce ambiguity regarding which spins participate in the measured chains, particularly when a spectral region in the chain contains multiple spins (see Fig. 1d).

We isolate the interaction of nuclear spins with the electron spin through an electron-nuclear double-resonance block acting at a selected nuclear-spin-frequency region. The key idea is that the frequency shift imprinted by the electron spin sensor can be recoupled by controlling the electron spin state. We use microwave pulses that transfer the electron population from the $|-1\rangle$ to the $|+1\rangle$ state (Fig. 3b, see "Methods" section). The nuclear spin is decoupled from quasi-static noise and the rest of the spins, extending its coherence time, while the interaction of interest ($\Delta_i$) is retained.

Figure 3 shows an example for a nuclear spin at $A_1$, for which we measure a hyperfine shift $\Delta_1 = 14549.91(5)$ Hz and a spectral linewidth of 1.8 Hz (Fig. 3d and e). Besides a tool to distinguish individual spins in the network with high spectral resolution, this method has the potential for improved characterization of the hyperfine interaction in electron-nuclear-spin systems.

The observed coherence time $T_2 = 0.36(2)$ s is slightly shorter than the bare nuclear-spin-echo time $T_{2,SE} = 0.62(5)$ s. This reduction is caused by a perturbative component of the hyperfine tensor in combination with the finite magnetic field strength (see Supplementary Note 4). Flipping the electron spin between $m_s = \pm 1$ changes the quantization axes of the nuclear spins, which causes a change of the nuclear-nuclear interactions[18], which is not decoupled

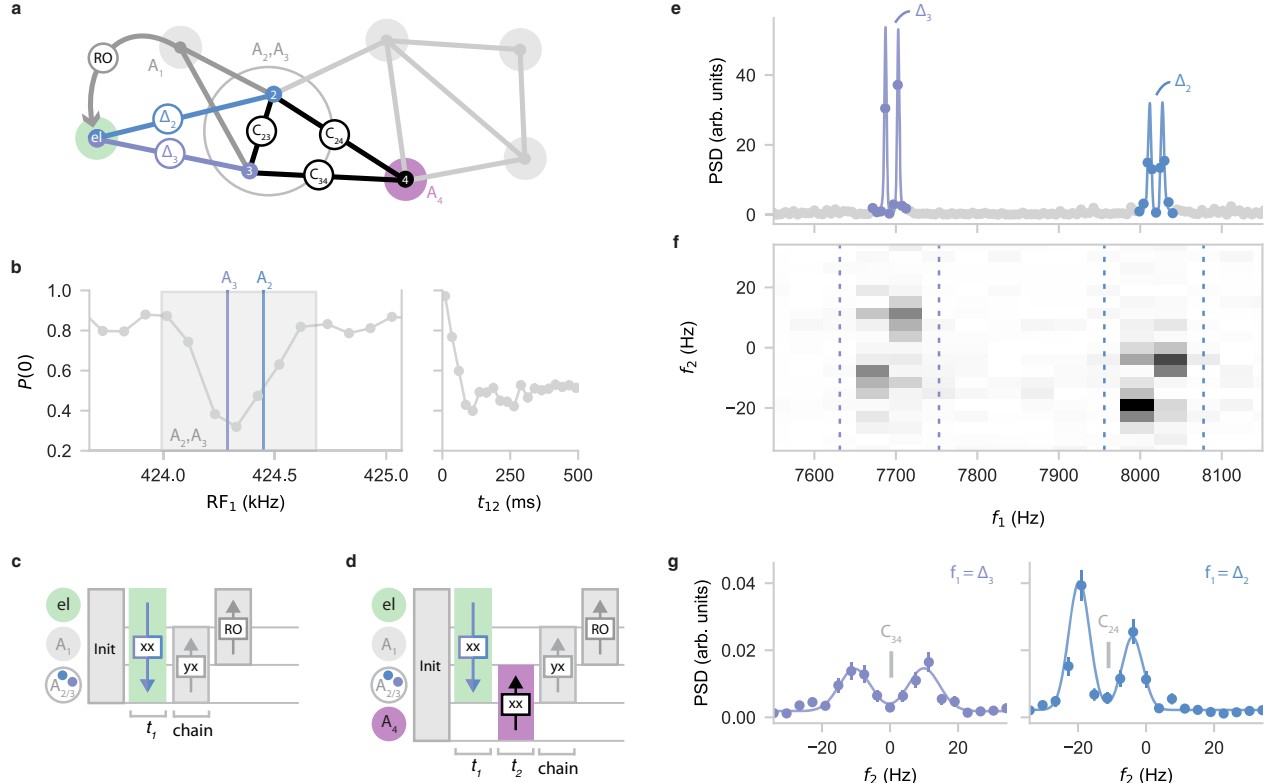

**Fig. 4 | Two-dimensional spectroscopy of spectrally crowded spins. a** Schematic of the studied system, which contains two spins with overlapping frequencies $A_2 \approx A_3$ (gray circle), with slightly different hyperfine shifts ($\Delta_2$, $\Delta_3$). Both are coupled to the spin at $A_1$, which is used for to transfer the signal to the electron spin for readout ('RO'). **b** SEDOR spectroscopy (as in Fig. 2d) of the frequency region $A_2$, $A_3$, with the estimated spin frequencies indicated. Sweeping the $t_{12}$ evolution time results in a quick decay. **c** Pulse sequence for the electron-nuclear double-resonance sequence used in (**e**), where the $\Delta_i$ are extracted by sweeping $t_1$. **d** Pulse sequence combining electron-nuclear and nuclear-nuclear double resonance, used in (**f**). Adding a nuclear-nuclear block (pink) and sweeping both $t_1$ and $t_2$ reveals the correlation between $\Delta_i$ and spin-spin couplings. **e** Sweeping $t_1$ yields a high-resolution PSD of the $A_2$, $A_3$ frequency region, showing two (split) frequencies $\Delta_2$ and $\Delta_3$. The solid curve is a four-frequency fit to the data. **f** Signal (PSD) for the two-dimensional sequence, revealing two distinct regions along the $f_1$-axis at $\Delta_2$ and $\Delta_3$. **g** Binned line cut of (**f**) along the $f_2$-axis at frequencies $\Delta_2$, $\Delta_3$ (region indicated by dotted lines). The positions of the (split) peaks indicate the coupling to the spin at $A_4$ ($C_{24} = -11.8(2)$ Hz, $C_{34} = -0.2(5)$ Hz). The solid line is a fit of two Gaussians to extract the couplings.

by the spin-echo sequence (see Supplementary Fig. 4). The effect is strongest for spins near the NV center. For larger fields or for spins with weak hyperfine couplings, we expect that further resolution enhancement is possible by applying multiple refocusing pulses (see Supplementary Note 4).

Finally, we combine spin-chain sensing and electron-nuclear double resonance to correlate high-resolution spin frequencies ($\Delta_i$) with specific spin-spin couplings ($C_{ij}$), even when a chain contains multiple spins with overlapping frequencies. We illustrate this scheme on a chain of spins, where two spins (2 and 3) have a similar frequency ($A_2 \approx A_3$) and both couple to $A_1$ and $A_4$ (Fig. 4a). The goal is to extract $\Delta_2$, $\Delta_3$ and the couplings to Spin 4 ($C_{24}$, $C_{34}$). As a reference, standard double-resonance shows a quickly decaying time-domain signal, indicating couplings to multiple spins that are spectrally unresolved (Fig. 4b).

Figure 4 c shows how the electron-nuclear double-resonance sequence (mint green) is inserted in the spin-chain sequence to perform high-resolution spectroscopy of the $A_2$, $A_3$ frequency region. Sweeping the interaction time $t_1$ shows multiple frequencies (Fig. 4e), hinting at the existence of multiple spins with approximate frequency $A_2$. The result is consistent with two spins at frequencies $\Delta_2 = 8019.5(2)$ Hz and $\Delta_3 = 7695.2(1)$ Hz, split by an internal coupling of $C_{23} = 7.6(1)$ Hz (Fig. 4a and Supplementary Fig. 2e, f).

Next, we add a nuclear-nuclear block (pink block in Fig. 4d) and sweep both electron-nuclear ($t_1$) and nuclear-nuclear ($t_2$) double-resonance times to correlate $\Delta_2$ and $\Delta_3$ with nuclear-nuclear couplings

$C_{24}$ and $C_{34}$. After the $t_1$ evolution, the hyperfine shifts $\Delta_i$ are imprinted in the z-expectation value of each spin, effectively modulating the nuclear-nuclear couplings observed in $t_2$. The 2D power spectral density (PSD) shows signals in two distinct frequency regions along the $f_1$-axis, corresponding to $\Delta_2$ and $\Delta_3$ (Fig. 4f). Analysing the nuclear-nuclear ($f_2$) signal at these frequencies (Fig. 4g), we find $C_{24} = -11.8(2)$ Hz and $C_{34} = -0.2(5)$ Hz. We attribute the splitting to the coupling $C_{23}$ between Spins 2 and 3 (see "Methods" section, Supplementary Fig. 2g, h). Varying $RF_4$ enables the measurement of the interactions of spins 2 and 3 to other parts of the network (for example to determine $C_{12}$, $C_{13}$). Beyond the examples shown here, the electron-nuclear block can be inserted at specific positions in the spin-chain sequence (Fig. 2c) to extract $\Delta_i$ of all spins in the chain (Supplementary Fig. 9).

## Reconstruction of a 50-spin network

Finally, we apply these methods to map a 50-spin network. The problem resembles a graph search (see "Methods" section)[42]. By identifying a number of spin chains in the system, and fusing them together based on overlapping sections, we reconstruct the connectivity (Fig. 5). Limited-sized chains are sufficient because the couplings are highly non-uniform, so that a few overlapping vertices and edges enable fusing chains with high confidence. We use a total of 249 measured interactions through pairwise and chained measurements. Fusing these together provides a hypothesis for the network connectivity (Fig. 5b).

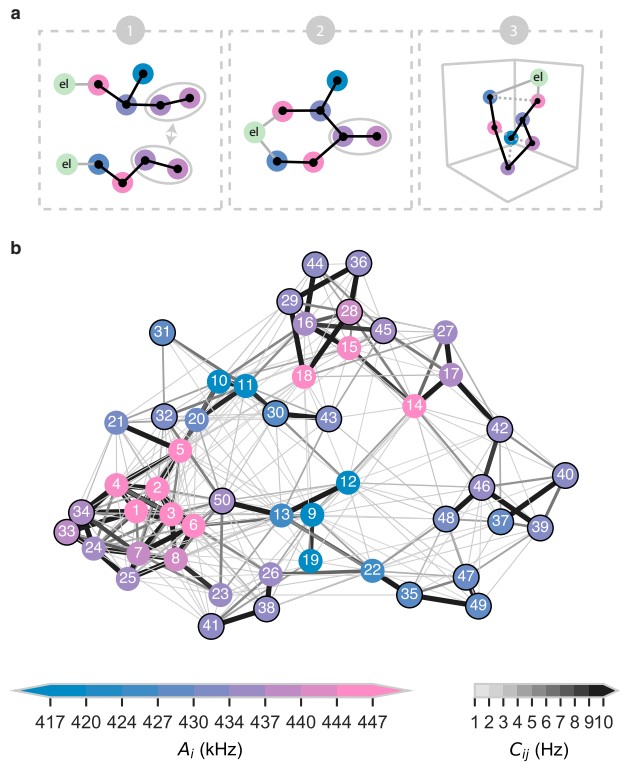

**Fig. 5 | Mapping a 50-spin network. a** Schematic illustrating the procedure for mapping large networks. 1: Separate high-resolution chains through the network are measured (two example chains shown here). 2: We merge chains that share a common section of the network. 3: Optionally, an algorithm adapted from Abobeih et al.[18] estimates the most likely spin positions (see "Methods" section), which predicts all unmeasured nuclear-nuclear couplings (dotted lines) and provides a validation for the assignment and merging of step 2. **b** Graph of the 50-spin network mapped in this work, with edges indicating spin-spin interactions above 2 Hz and vertex colors denoting spin frequencies $A_i$. The spins are labeled according to Supplementary Table 1. Black circles indicate the 23 newly mapped spins compared to previous work[18]. A 3D spatial image of the network is presented in Supplementary Fig. 3.

To validate our solution for the network we use the additional information that the nuclear-nuclear couplings can be modeled as dipolar and attempt to reconstruct the spatial distribution of the spins. Compared to work based on pairwise measurements[18], our spin-chain measurements provide additional information on the connectivity and coupling signs, reducing the complexity of the numerical reconstruction. Additionally, we constrain the position using the measured hyperfine shift $\Delta_i$ (see "Methods" section). Because the problem is highly overdetermined[18], the fact that a spatial solution is found that closely matches the measured frequencies and assignments validates the obtained network connectivity. Additionally, the reconstruction yields a spatial image of the spin network and predicts the remaining unmeasured 976 spin-spin interactions, most of which are weak (<1 Hz). An overview of the complete 50-spin cluster, characterized by 50-spin frequencies and 1225 spin-spin couplings can be found in Supplementary Table 1 and in Fig. 5b.

## Discussion

In conclusion, we developed correlated double-resonance sensing that can map the structure of large networks of coupled spins, with high spectral resolution. We applied these methods to reconstruct a 50-spin network in the vicinity of an NV center in diamond. The methods can be applied to a variety of systems in different platforms, including electron-electron spin networks[7-15,43]. Mapping larger spin systems

might be in reach using machine-learning-enhanced protocols and sparse or adaptive sampling techniques, which can further reduce acquisition times[44,45]. Combined with control fields[1,16,32], the methods developed here provide a basis for universal quantum control and readout of the network, which has applications in quantum simulations of many-body physics[1]. Furthermore, the precise characterization of a 50-spin network provides new opportunities for optimizing quantum control gates in spin-qubit registers[16,32,36], for testing theoretical predictions for defect spin systems[46], and for studying coherence of solid-state spins on the microscopic level, including quantitative tests of open quantum systems and approximations of the central spin model[47]. Finally, these results might inspire high-resolution nano-MRI of quantum materials and biologically relevant samples outside the host crystal.

## Methods
### Sample and setup

All experiments are performed on a naturally occurring NV center at a temperature of 3.7 K (Montana S50 Cryostation), using a home-built confocal microscopy setup. The diamond sample was homo-epitaxially grown using chemical vapor deposition and cleaved along the ⟨111⟩ crystal direction (Element Six). The sample has a natural abundance of $^{13}C$ (1.1%). The NV center has been selected on the absence of couplings to $^{13}C$ stronger than ≈ 500 kHz. No selection was made on other properties of the $^{13}C$ nuclei distribution. A solid immersion lens (SIL) that enhances photon collection efficiency is fabricated around the NV center. A gold stripline is deposited close to the edge of the SIL for applying microwave (MW) and radio-frequency (RF) pulses. An external magnetic field of $B_z = 403.553$ G is applied along the symmetry axis of the NV center, using a (temperature-stabilized) permanent neodymium magnet mounted on a piezo stage outside the cryostat[16]. The field is aligned to within 0.1 degrees using a thermal echo sequence[18].

### Electron and nuclear spins

The sample was previously characterized in Abobeih et al.[18] and the 27 nuclear spins imaged in that work are a subset of the 50 nuclear-spin network presented here. The NV electron spin has a dephasing time of $T_2^* = 4.9(2)$ μs, a Hahn spin-echo time of $T_2 = 1.182(5)$ ms, and a relaxation time of $T_1 > 1$ h[18]. The spin state is initialized via spin-pumping and read out in a single shot through spin-selective resonant excitation, with fidelities $F_0 = 89.3(2)$ ($F_1 = 98.2(1)$) for the $m_s = 0$ ($m_s = -1$) state, resulting in an average fidelity of $F_{avg} = 0.938(2)$. The readout is corrected for these numbers to obtain a best estimate of the electronic spin state. The nuclear spins have typical dephasing times of $T_2^* = 5–10$ ms and Hahn-echo $T_2$, up to 0.77(4) s[16]. $T_2$-times for spins with frequencies closer to the nuclear Larmor frequency ($\Delta_i \lesssim 5$ kHz) typically decrease to below 100 ms (see e.g. Fig. 2g, right panel), as the spin-echo simultaneously drives other nuclear spins at these frequencies which are recoupled to the target (instantaneous diffusion).

### Pulse sequences

We drive the electronic $m_s = 0 \leftrightarrow m_s = -1$ ($m_s = 0 \leftrightarrow m_s = +1$) spin transitions at 1.746666 (4.008650) GHz with Hermite-shaped pulses. For transferring the electron population from the $m_s = -1$ to the $m_s = +1$ state (Figs. 3 and 4), we apply two consecutive π-pulses at the two MW transitions, spaced by a waiting time of 3 μs. For all experiments, we apply RF pulses with an error-function envelope in the frequency range 400–500 kHz. Details on the electronics to generate these pulses can be found in ref. 1.

For most experiments described in this work, the measurable signal is dependent on the degree of nuclear-spin polarization. We use a dynamical-nuclear polarization sequence, PulsePol, to transfer polarization from the electron spin to the nuclear-spin bath[1,40]. The

number of repetitions of the sequence is dependent on the specific polarization dynamics of the spins being used in the given experiment but ranges from 500–10,000. The PulsePol sequence is indicated by the 'Init' block in the sequence schematics. All double-resonance sequences follow the convention illustrated in the dotted boxes in Fig. 2b, c, and Fig. 3b, where the horizontal gray lines denote different RF frequencies and the top line the electronic MW frequency. The two letters in the double-resonance blocks ('xx' or 'yx') denote the rotation axes of the first and final $\pi/2$-pulses. The $\pi$-pulses (along the x-axis) are applied sequentially (following ref. 18). The lengths of all RF pulses are taken into account for calculating the total evolution time. Nuclear spins are read out via the electron by phase-sensitive ('yx') dynamical decoupling; DD or DDRF sequences[16], indicated by the 'RO'-marked block in the sequence schematics. Typically, the spin that is read out with the electron is reinitialized via a SWAP gate before the final SEDOR block in order to maximize its polarization. However, all experiments presented here can be performed by using just the DNP initialization, albeit with a slightly lower signal-to-noise ratio.

## 2D spectroscopy experiments

For the 2D measurement, we concatenate an electron-nuclear double resonance with a nuclear-nuclear SEDOR. For every $t_1$-point, we acquire 20 $t_2$ points, ranging from 10 to 260 ms. The final $\pi/2$-pulse of the electron double resonance and the first of the SEDOR are not executed, as they can be compiled away. To correct for any slow magnetic field drifts that lead to miscalibration of the two-qubit gate used for readout, causing a small offset in the measured signal, we set our signal baseline to the mean of the final five points ($\approx$200–260 ms), where we expect the signal to be mostly decayed. Note that these field drifts do not affect any of the double-resonance blocks in which the quantities to be measured are encoded (due to the spin-echo).

Both the 1D (Fig. 4e) and 2D (Fig. 4f) signals are undersampled to reduce the required bandwidth. To extract $\Delta_2$, $\Delta_3$, we fit a sum of cosines to the time-domain signal of Fig. 4e. To extract the frequencies along the $f_2$-axis, which encode the nuclear-nuclear couplings ($C_{24}$, $C_{34}$), we take an (extended) line cut at $f_1 = \Delta_2$ and $f_1 = \Delta_3$. To increase the signal, we sum over the four bins indicated by the dotted lines. We fit two independent Gaussians to the $f_2$-data to extract $C_{24}$ and $C_{34}$. We find splittings of 7.8(2) Hz and 10.2(5) Hz, respectively, whose deviation with respect to measurements in Fig. 4e is unexplained. The skewed configuration of the two peaks (lower left, upper right) is a result of the correlation of the neighboring spin state between the $t_1$ and $t_2$ evolution times. The different ratio of signal amplitudes belonging to Spin 2 and Spin 3, between the 1D and 2D electron-nuclear measurements are due to using different settings for the chained readout (evolution time, RF power). As we are only interested in extracting frequencies, we can tolerate such deviations.

Supplementary Fig. 2 shows numerical simulations of the experiments presented in Fig. 4. These are generated by evaluating the Hamiltonian in Supplementary Eq. 3, taking into account the two spins at $A_2$, $A_3$, the spin at $A_4$, and the electron spin.

## Network reconstruction

Here, we outline a general procedure for mapping the network by performing specific spin-chain and high-resolution $\Delta_i$ measurements. The mapping-problem resembles a graph search, with the NV electron spin used as root[42]. We base the protocol on a breadth-first-like search, which yields a spanning tree as output, completely characterizing the network. The following pseudocode describes the protocol:

**Algorithm 1.** *Input*: physical spin network, initial vertex *el*
 *Output*: breadth-first tree *T* from root *el*
 $V_0 = \{el\}$ ▷ Make *el* the root of *T*, $V_i$ denotes the set of vertices at distance *i*
 $i = 0$

 **while** $V_i \neq \emptyset$ **do** ▷ Continue until network is exhausted
 **for** each vertex $v \in V_i$ **do**
 **for** each frequency *f* **do**
 C, singlecoupling = MeasureCoupling($v,f$) ▷ Returns coupling *C* between vertex *v* and frequency *f*
 **if** singlecoupling **then** ▷ Checks if MeasureCoupling returned a single, resolvable coupling
 create vertex *w*
 $A_w = f$
 $C_{vw} = C$
 unique, duplicate = CheckVertex($w, T$) ▷ Checks if *w* was already mapped in *T*
 **if** unique **then** ▷ *w* was not yet mapped
 add *w* to $V_{i+1}$ in *T* ▷ *w* is added to *T* as a new vertex
 **end if**
 **if not** unique **and** duplicate == *k* **then** ▷ *w* is the same vertex as *k* in *T*
 add $C_{vk} = C_{vw}$ in *T* ▷ The measured coupling is assigned to *k*
 delete *w*
 **end if**
 **if not** unique **and** duplicate == None **then** ▷ Undecided if spin was mapped
 delete *w* ▷ *w* is not added to *T*
 **end if**
 **end if**
 **end for**
 **end for**
 $i = i + 1$
 **end while**

New vertices that are detected by chained measurements are iteratively added, once we verify that a vertex was not characterized before (i.e. has a duplicate in the spanning tree *T*). The function MeasureCoupling($v, f$) performs a spin-echo double-resonance sequence between vertex *v* and a frequency *f*, (a spin chain of length $i-1$ is used to access *v*) and checks whether a single, resolvable coupling is present (stored in the boolean variable 'singlecoupling'). In the case that *v* is the electron spin (*el*) an electron-nuclear DD(RF) sequence is performed[16,41]. The function CheckVertex($w, T$) instructs the experimenter to perform a number of spin chain and electron-nuclear double-resonance measurements, comparing the vertex *w* and its position in the network with that of the (possibly duplicate) vertex *k* (see Supplementary Note 5). If one of these measurements is not consistent with our knowledge of *k*, we conclude *w* is a unique vertex and add it to *T*. If all measurements coincide with our knowledge of *k*, we conclude it is the same vertex and merge *w* and *k*. If the CheckVertex($w, T$) is inconclusive (e.g. due to limited measurement resolution), we do not add *w* to *T*. Note that the measurement resolution, determined by the nuclear $T_2$-time, is expected to decrease for spins further away from the NV center (See Supplementary Note 2). This eventually limits the number of unique spins that can be identified and added to the network map.

The platform-independent procedure outlined above can be complemented by logic based on the 3D spatial structure of the system[18]. For example, when the CheckVertex($w, T$) function is inconclusive, one can sometimes still conclude that *w* must be unique (or vice versa equal to *k*), based on the restricted number of possible physical positions of these two spins in 3D space[18]. In practice, we alternate the graph search procedure with calls to a positioning algorithm[18], which continuously checks whether the spanning tree *T* is physical and aides in the identification of possible duplicates.

## 3D spatial image

For the 3D reconstruction of the network, we use the positioning algorithm developed in ref. 18. To limit the experimental time we re-

use the data of ref. [18] and add the new measurements to it in an iterative way. We set the tolerance for the difference between measured and calculated couplings to 1 Hz. Although we only measure the new spin-spin couplings and chains when the electron is in the $m_s = -1$ state, we can assume this is within tolerance to the average value of the coupling if the perpendicular hyperfine component is small (<10 kHz)[18]. The spin positions are restricted by the diamond lattice. Spins that belong to the same chain are always added in the same iteration and up to 10000 possible configurations are kept. Chains starting from different parts of the known cluster can be positioned in a parallel fashion if they share no spins, reducing computational time. For spins that are relatively far away from the NV, we also make use of the interaction with the electron spin, approximating the hyperfine shift $\Delta_i$ to be of dipolar form within a tolerance of 1 kHz (neglecting the Fermi contact term[46]). For those cases, we model the electron spin as a point dipole with origin at the center of mass, as computed by density functional theory[46]. If multiple solutions are found, we report the standard deviation of the possible solutions as a measure of the spatial uncertainty (see Supplementary Table 1).

### Error model and fitting
Confidence intervals assume the measurement of the electron state is limited by photon shot noise. The shot-noise-limited model is propagated in an absolute sense, meaning the uncertainty on fit parameters is not rescaled to match the sample variance of the residuals after the fit. For all quoted numbers, the number between brackets indicates one standard deviation or error indicated by the fitting procedure. We calculate the error on the PSD according to ref. [48], assuming normally distributed errors.

## Data availability
All data underlying the study are available on the open 4TU data server under accession code: https://doi.org/10.4121/aba1cc84-0aea-4cdc-93ca-68b0db38bd81.v1.

## Code availability
Code used to operate the experiments is available on request.

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

## Acknowledgements

We thank V. V. Dobrovitski for useful discussions, and H.P. Bartling and S.J.H. Loenen for experimental assistance. This work is part of the research program NWA-ORC (NWA.1160.18.208 and NWA.1292.19.194), (partly) financed by the Dutch Research Council (NWO). This work was supported by the Dutch National Growth Fund (NGF), as part of the Quantum Delta NL program. This work was supported by the Netherlands Organisation for Scientific Research (NWO/OCW) through a Vidi grant. This project has received funding from the European Research Council (ERC) under the European Union's Horizon 2020 research and innovation program (grant agreement No. 852410). This work was supported by the Netherlands Organization for Scientific Research (NWO/OCW), as part of the Quantum Software Consortium Program under Project 024.003.037/3368. S.A.B. and L.C.B. acknowledge support from the National Science Foundation under grant ECCS-1842655, and from the Institute of International Education Graduate International Research Experiences (IIE-GIRE) Scholarship. S.A.B. acknowledges support from an IBM PhD Fellowship.

## Author contributions

GLvdS, DK, and THT devised the experiments. GLvdS performed the experiments and collected the data. CEB, JR, SAB, LCB, MHA, and THT performed and analyzed preliminary experiments. GLvdS, DK, CEB, and JR prepared the experimental apparatus. GLvdS, DK, and THT analyzed the data. MM and DJT grew the diamond sample. GLvdS and THT wrote the manuscript with input from all authors. THT supervised the project.

## Competing interests

T.H. Taminiau, G.L. van de Stolpe, C.E. Bradley, J. Randall, and D.P. Kwiatkowski declare competing interests in the form of a Dutch patent application. Patent applicant: Technische Universiteit Delft. Name of inventor(s): Taminiau, Tim Hugo; van de Stolpe, Guido Luuk; Bradley, Conor Eliot; Randall, Joe; Kwiatkowski, Damian Patryk. Application number: NL2035279. Status of application: filed, waiting for search report. Covered aspects: Full content of manuscript. M.H. Abobeih, S.A. Breitweiser, L.C. Basset, M. Markham, and D.J. Twitchen declare no competing interests.
