## [Peer Review File · Nature Communications]

REVIEWERS' COMMENTS

Reviewer #1 (Remarks to the Author):

This is a milestone in the research of quantum sensing of nuclear spin clusters. Using double and multiple resonance control of a central electron spin and nuclear spins along a chain, the authors developed a method to re-construct the frequencies and the neighboring couplings of the nuclear spins in a chain, in a cascaded manner. The resolution of the frequencies is only limited by the T2 coherence time (excluding the effects of static noises). By fusing multiple nuclear spin chains obtained using this method, the authors managed to reconstruct the interaction network structure of 50 nuclear spins (so far the record) and, assisted by numerical fitting, mapped the spatial structure of the network (which can give many other unmeasured parameters of the dipole-dipole interactions between the nuclear spins). This method can be generalized to other spin systems and in other materials. As pointed out by the authors, it may even be applied to determining the nuclear spin spatial configurations in large molecules outside the diamond sample. The nuclear spin network mapped out by this work can be a stage for quantum simulation of quantum many-body physics in irregular quantum networks. The paper is clearly written, the data and analysis are convincing. I strongly recommend rapid publication of the paper.

One optional revision is recommended: The authors may add a Supplementary table showing the coordinates of the 50 spins, so that interested readers could play some numerical experiments using the network, which could increase the impact of this experimental work.

Reviewer #2 (Remarks to the Author):

The manuscript by van de Stolpe et al. is impressive for many reasons. It is extremely thorough, well-communicated, and impactful. In short, I think Nature Communications is very fortunate to have the opportunity to publish this paper.

The authors use a combination of correlated sensing, double spin resonance techniques, and use of spin chains to measure and map the largest cluster of nuclear spins in a solid to date. This is achieved by coupling to a central electron spin qubit that can be read-out optically with high fidelity at low temperature. This work has important implications for quantum communications (more nuclear registers), computations (author's related works on error correction, more qubits), and sensing— for example eventually to nuclei external to the solid-state material (the end goal for many groups

studying this system). The author's advance almost doubles the cluster size that this group demonstrated in Nature in 2019 (27 spins). Importantly, this work doesn't just brute force the experiment to make the cluster bigger— they introduce new schemes and understanding to provide new and extended functionality.

For example, an outstanding problem for nuclear sensing is spectral and spatial crowding— such that nuclear spins cannot be uniquely identified or controlled. This work overcomes that limitation with a suite of double resonance techniques, some of which are nicely sign sensitive, that can be chained together to extend the network of interactions and elucidate the connectivity (over 1225 spin-spin interactions mapped). Overall, the author's work is well explained, the figures are clear and the supplement impressive and useful. I especially enjoyed the "arrow" nomenclature throughout the manuscript (Starting in Fig. 2b) which made the scheme clear. This manuscript also performs a spatial mapping/estimate of the full 50 qubit structure. I will say that reading the SI of this paper made me think "awesome" more than a couple of times. All of this speaks to the quality and depth of this work.

The authors also correctly note a relevant preprint (Ref 45), which is very similar. However, this work goes drastically beyond Ref 45 experimentally, which only measured the "first shell" of nuclear spins, and in terms of the theoretical/practical understanding.

While I think this manuscript could be published as-is. I have a few questions and suggestions to improve clarity.

-In figure 1, the dotted outlines and "a,b,c" labels are not very visible/too small

-Some mention (Perhaps in the SI) should be made on the limit of the spin chain approach. How much longer could the chains be, how big a cluster? What are the prospects for the future?

-For example, why are chains limited to 5? Is there some fidelity/polarization transfer infidelity? Can the authors quantify?

-A quick statement in the SI might be nice about why certain nuclear spins are polarized with PulsePol vs measurement/swap.

-In this work, the nuclear bath is polarized, freezing many interactions. Could similar schemes work with unpolarized baths? Understanding that coherences and flip-flops hurt.

-The discussion starting line 220 is a little unclear. Would consider rewriting.

-I am curious how this scheme works when extending beyond just sensing the presence of the spins. Can similar sequences be used to utilize these distant spins for quantum applications? (perform gates?)

-Line 244 could use a reference for t_2^* (SI?)

-How does measurement of delta reduce ambiguity exactly? Line 248.

-Sentence starting line 250 unclear.

-There is a question on how certain the chains are reconstructed, and how uniquely spins are identified. What are the cases where this scheme can be confused/breaks down? Which spins can it not access?

-Related to my other question. How far out (spatially) might one expect to extend the sensing? What about isotopically purified samples?

-Why are only two spin chains measured? It seems some nuclei are missing? I recommend on Figure 5B to label/show the newly accessed spins compared to previous work.

-Why was DDRF not used?

Reviewer #3 (Remarks to the Author):

Dear editors,

Thank you for this opportunity to review the manuscript “Mapping a 50-spin-qubit network through correlated sensing.” The manuscript reports on double-resonance methods to measure the single-spin and two-spin coefficients of a (classical) Ising spin network. The authors experimentally demonstrate these methods by performing experiments on a nitrogen-vacancy center in diamond that is coupled to an ensemble of nearby nuclear spins. The experimental methods used are not per se novel, but the demonstration of sequential two-spin correlation blocks on chains of multiple spins and their deployment at scale in a real sample is technically impressive and non trivial. There is also novelty in how the data is processed (Sec. E). A another key achievement is extending their previous results (Nature 576, 411–415 (2019)) from 27 spins to 50 spins, offering opportunities for realizing quantum information processing protocols using solid-state spin qubits.

My main concern is that the manuscript and the figures are very difficult to read and to parse. I believe that it will be hard for scientists from other communities or those that are not immediately familiar with the previous work of the authors to understand the work and make use of these methods in other context. The manuscript includes many instances where greater specificity in

language choice, proper definitions, and pedestrian explanations would help the reader better understand the work and its subtleties.

For these reasons, I suggest having the authors further revise their text and figures to improve the clarity of their manuscript before making a final decision on publication. I foresee that a significantly edited manuscript would be suitable for publication in Nature Communications.

General comments:

- Figures could be significantly edited to improve knowledge transfer to the reader (see comments below). Figures are generally too crowded (in both space and colors), hard to parse due to slight inconsistencies between main text, captions, and figures, and quite technical.
- Please revise your labeling and/or network representation conventions to insure consistency between your Hamiltonian and the network representation. I suggest independently labeling spectral classes and spin within each spectral classes, as well as avoiding the polychromatic representation of spectral classes. There are too many colors used for different purposes: color is meaningfully used to isolate a chain of spins within the network; however, Fig. 1 uses dark-purple for nodes, whereas Fig. 2 uses multicolor nodes. This coloring seems inconsistent/redundant as spectral classes are already labeled by the A_j coefficients.
- Please revise the text so that all technical terms are properly defined, especially those that have ambiguous meanings, and that no jargon/loose terms are used. Whenever possible, add short explanations on what is implicitly assumed because it is common knowledge in your community, but might not be shared knowledge by all readers, e.g., describing in a few sentences how the SEDOR sequence gains information about the interaction strength between two spins, or how “the nuclear-spin state is mapped as a polarization difference on the NV electron spin, which is readout through spin-dependent fluorescent in a confocal microscope operating at 3.7 K (see Methods).” These short sentences help anchoring the reader on known facts without having them to seek information elsewhere.

Abstract

- “of current methods”: what methods are you referring to?
- “complexity of these spin networks”: “complexity of a network” is ambiguous, please revise. I assume that by “size” you mean the number of spins/nodes, and by “complexity” the number of edges. Also, how is “complexity” constrained by spatial resolution?
- “through high-resolution correlated sensing schemes”: the usage of “correlated sensing” is ambiguous, as it is not clear what is correlated with what (see below for further discussion); consider using a different term or being more specific.

- “with high spectral resolution”: the usage of “high” is ambiguous and relative; rather emphasize that your methods are improving the spectral resolution over previously demonstrated methods; also specify what increase in size your methods enable, e.g., going from 27 spins to 50 spins.
- “opportunities for quantum simulation”: be more specific to make the statement more convincing as accessing more spins without proper control and interactions does not necessarily lead to greater simulation capability, and classical spin Hamiltonian (1) does not allow for universal quantum simulation, e.g., “opportunities for simulating spin dynamics in Ising spin networks” (same comments for references about quantum simulations in the introduction).

Introduction

- “a key open challenge is spectral crowding”: given that the abstract states that the problem is “spectral resolution of current methods”, the root cause of “spectral crowding” should be better explained. My point is that spectral crowding seems to depend on both the “spectral resolution of current methods” being limited AND the distribution of spectral contributions in the sample. Because the number of spins with the same frequency increases with the distance away from the central spin (the volume of a spherical shell approximately scales as r^2), spectral crowding seems to be more likely for far-away/low-frequency spins. Thus, spectral crowding depends on both resolution and bandwidth of detection method (smallest frequency that can be accessed), which in turn depends on the density of the sample. If this statement is true, because “spectral crowding” depends on sample properties, to access large system size for “quantum simulation”, you might then choose to either increase the density of nuclear-spin defects in the sample (because nuclear spins close to the NV are less likely to spectrally overlap) or access nuclear-spin defects at greater distance. It is not clear if your method performs equally well for the two cases. If your approach depends on the density of nuclear spins in the sample, then the importance of your results would be less about the total number of spins (50 spins) than the total number of spins in a given detection volume (the density). In short, some explanation is lacking for the reader to understand the relationships between spectral resolution, spectral crowding, and network size (some of it is implied, but not explicitly addressed).
- as mentioned above, “network connectivity” is ambiguous, as it might refer to whether a vertex is connected with another one, which is always true for all-to-all coupled spin networks. Please be more specific, e.g., “that measure the frequencies and couplings of a network of spins”.
- “measure chains of coupled spins”: it is not clear what you mean by “measuring chains”. Please be more specific when describing the key concept of your method. Also, “these spin chains remove ambiguity” is ambiguous, as “spin chains” is not the proper subject of the verb “remove”. (Generally, look for similar instances of loose/ambiguous language).

Section A:

- The sum Eq. (1) runs up to $N/2$, which is ambiguous for N odd. Consider using $\sum_{i=1}^N$ $\sum_{j=i+1}^N$.

- Eq. (1) is a priori valid for any Ising spin network, including both electron and nuclear spin defects. Specify whether results presenting in the manuscript are generally valid for all spins or only applicable to nuclear spins (even though the demonstration is made for nuclear spins).

- Under which conditions is Eq. (1) valid?

- "In general, the number of frequency vertices is smaller than the number of spins, as multiple spins might occupy the same frequency vertex." A distinction would ideally be made between the true underlying physical all-to-all spin network representing Hamiltonian (1) (for which all coefficients are non-zero) and the estimated network reconstructed from measurements with finite resolution (where some coefficients might be indistinguishable from zero or overlapping). For example, one node in Fig. 1 is labeled as A_23 which is very confusing, as it does not match definitions in Eq. (1).

Paragraph 84-92:

- "State-of-the-art" demonstrations: it is quite interesting that the authors refer to their previous work as "state-of-the-art" in plural form. It be more appropriate to use "In our previous work" or "In our more recent work".

- Please add references to further expand on existing approaches to spin network matpping, as well as prior uses of SEDOR-like sequences, possibly including prior work with NV centers for both electron and nuclear spins, as well as prior work in NMR.

- Please provide enough information for the reader to understand your previous work without having to access and read your previous paper or Methods section. E.g., in this paragraph reviewing previous work and describing its limitations, it is not clear how the NV center is involved and whether these "SEDOR" sequences are "correlated sequences" in the way that was defined in the abstract. It is also not clear if by " T_2, T_2^* ", you are referring to the decoherence time of the NV electron spin or of the nuclear spins (5 ms is mentioned in Sec. D)? If the decoherence time of the nuclear spins, does it change among spins, what limits its value, what is its (typical) value in your current system? What is its associated spectral resolution?

- Consider adding a clear upfront statement about what was done in your previous work and what is new with this work.

Line 133: "Correlated": What do you mean by "correlated"? "Correlated sensing" is mentioned multiple times in passing without a clear definition. I understand that the recoupled spin-echo sequence correlates the spin states of two spins by mapping the operator $I_z * I_d$ to $I_z * I_z$ with a modulation factor that depends on the interaction strength; however, "correlated" might be understood differently by different readers, cf. correlation spectroscopy of Laraoui2013. Please make sure that jargon is properly defined and its usage justified.

Line 145: "This allows for the merging of chains": What do you mean by "the merging of chains."
How is this done in practice?

Section E: Does the main text has an explicit reference to Fig. 4e? Results reported in Fig. 4d-f seems very interesting, but it is hard to understand them as the main text offers too little explanation.

Figure 1

This figure would benefit from further editing:

- The fully-connected network is confusing because many nodes have the same labels (e.g., A_1). The (a-b-c-d) labels are also hidden in the network and not aligned. I suggest having 4 disjoint sub-figures with the (a-b-c-d) labels in black at the upper left corner (following typical APS/Nature publication standards) presenting different networks or emphasizing different features of the same network.

- The vertex represented as purple disks represent "spin precession frequencies", but the edges connect spins.

- There are too many colors that are hard to distinguish from one another.

- "Mapping complex spin networks." I do not understand the usage of complex, as what is shown is a standard spin network.

(a) - "one spin in each vertex": this nomenclature is confusing, as it does not directly connects with the Hamiltonian from Eq. (1). An easy fix would be to revise Eq. (1) to include subindex, e.g., $A_{\{jk\}}$ labeling spectral classes, and spins within each spectral classes; however, this choice might be confusing when considering the $C_{\{jk\}}$ which are specified for spins rather than spectral classes.

(b) In Fig. 1(b), you might have the two A_2 spins labeled as $A_{\{2,1\}}$ and $A_{\{2,2\}}$ upon lifting the degeneracy. Also, if the blue spin was connected to the purple spin then $C_{\{2,4\}}$ would be defined twice, which is very confusing.

- "By measuring chains": this language is once again ambiguous

(c) The graph shows $C_{\{1,4\}}$ with an X, but the caption says "the coupling $C_{\{34\}}$ is not directly accessible from the spin $A_{\{1\}}$." Why id XX used first, then XY? What if XX was used instead of XX?

Figure 2

- this figure is very crowded and the different colors are overwhelming. Please simplify.

- "coloured spheres" -> "coloured disks"

- to be consistent with Fig. 1, A_j labels should be associated with spectral classes rather than spins

- Because there is only spin per spectral class, it might be sufficient to have labeled nodes rather than dots within colored nodes.

- I suggest removing the C_{jk} from circles and having them labeling the edges.

(b) What does “RO” refer to? How is “t” in “XX” and “RO” chosen? Could it be that t_{12} refers to the time of the correlation block applied to spin 1-2? Is “t” fixed or varying? A variable might be denoted by \vec{t} . Why is there an arrow pointing up?

- colors of circles make it hard to read the labels.

(c) “correlating 5 spin frequencies and 4 spin-spin interactions”: the sequence correlates the spin states of 5 spins; how does it correlate frequencies and interactions? Please be specific in your language to avoid introducing confusion.

“To probe the connection at “A4”? What does that mean?

(d-g) It would be beneficial for the reader to describe how the sequence is deployed to extract the relevant coefficients, e.g. “We sweep the frequency of the recoupling pulse on the second spin to extract its resonance frequency. We then sweep the time of the recoupling sequence to extract the coupling strength between the first and second spin. We repeat these steps for all pairs of spins in the chains.”

Figure 3

(a) What is “RO”? Why is there a need to represent the hyperfine interaction by the blue dotted line? To be consistent with the main text, do you mean the “nuclear-spin frequencies ω_i ”? “Between the nuclear spin and the electron” -> “Between the nuclear spin and the electron spin.”

(b) “Picking up signal from the interaction” is too vague to be meaningful. The diagram would ideally indicate a frequency/detuning to connect with (c). Is the electron or nuclear-spin frequency being swept?

(d) What is the coherence time of 0.36(2) s limited by? Is that the T_2^* of the nuclear spin?

(e) “that is ~ 75 times improved”: with respect to what reference?

Figure 4

- How is a spin cluster defined?

- Revise A_{23} to be consistent with Eq. (1)

- “chained readout”: it is a nice term, but ambiguous: chained readout of what?

- “of the frequency region A_{23} ”? Is A_{23} a frequency or a frequency region?

- “reveals the correlation between Δ_i and spin-spin couplings”: the main text makes a similar statement in Line 283, but it is done too fast without a clear explanation of why Δ_i is correlated with spin-spin couplings.

- More explanation is needed to understand Fig. 4e.

Reviewer #1

This is a milestone in the research of quantum sensing of nuclear spin clusters. Using double and multiple resonance control of a central electron spin and nuclear spins along a chain, the authors developed a method to re-construct the frequencies and the neighboring couplings of the nuclear spins in a chain, in a cascaded manner. The resolution of the frequencies is only limited by the T2 coherence time (excluding the effects of static noises). By fusing multiple nuclear spin chains obtained using this method, the authors managed to reconstruct the interaction network structure of 50 nuclear spins (so far the record) and, assisted by numerical fitting, mapped the spatial structure of the network (which can give many other unmeasured parameters of the dipole-dipole interactions between the nuclear spins). This method can be generalized to other spin systems and in other materials. As pointed out by the authors, it may even be applied to determining the nuclear spin spatial configurations in large molecules outside the diamond sample. The nuclear spin network mapped out by this work can be a stage for quantum simulation of quantum many-body physics in irregular quantum networks. The paper is clearly written, the data and analysis are convincing. I strongly recommend rapid publication of the paper.

We thank the referee for their recommendation and feedback.

1. One optional revision is recommended: The authors may add a Supplementary table showing the coordinates of the 50 spins, so that interested readers could play some numerical experiments using the network, which could increase the impact of this experimental work.

This information is presented in Supplementary table I. We now added an explicit reference in the last paragraph of the 'reconstruction of a 50-spin network' section.

Reviewer #2

The manuscript by van de Stolpe et al. is impressive for many reasons. It is extremely thorough, well-communicated, and impactful. In short, I think Nature Communications is very fortunate to have the opportunity to publish this paper.

The authors use a combination of correlated sensing, double spin resonance techniques, and use of spin chains to measure and map the largest cluster of nuclear spins in a solid to date. This is achieved by coupling to a central electron spin qubit that can be read-out optically with high fidelity at low temperature. This work has important implications for quantum communications (more nuclear registers), computations (author's related works on error correction, more qubits), and sensing— for example eventually to nuclei external to the solid-state material (the end goal for many groups studying this system). The author's advance almost doubles the cluster size that this group demonstrated in Nature in 2019 (27 spins). Importantly, this work doesn't just brute force the experiment to make the cluster bigger— they introduce new schemes and understanding to provide new and extended functionality.

For example, an outstanding problem for nuclear sensing is spectral and spatial crowding— such that nuclear spins cannot be uniquely identified or controlled. This work overcomes that limitation with a suite of double resonance techniques, some of which are nicely sign sensitive, that can be chained together to extend the network of interactions and elucidate the connectivity (over 1225 spin-spin interactions mapped). Overall, the author's work is well explained, the figures are clear and the supplement impressive and useful. I especially enjoyed the "arrow" nomenclature throughout the

manuscript (Starting in Fig. 2b) which made the scheme clear. This manuscript also performs a spatial mapping/estimate of the full 50 qubit structure. I will say that reading the SI of this paper made me think “awesome” more than a couple of times. All of this speaks to the quality and depth of this work.

The authors also correctly note a relevant preprint (Ref 45), which is very similar. However, this work goes drastically beyond Ref 45 experimentally, which only measured the “first shell” of nuclear spins, and in terms of the theoretical/practical understanding.

While I think this manuscript could be published as-is. I have a few questions and suggestions to improve clarity.

We thank the referee for their recommendation for publication and their comments.

1. In figure 1, the dotted outlines and “a,b,c” labels are not very visible/too small

We reformatted the a,b,c,d labels according to Nature publication standards and to improve clarity (see also reviewer 3, comment 23).

2. Some mention (Perhaps in the SI) should be made on the limit of the spin chain approach. How much longer could the chains be, how big a cluster? What are the prospects for the future?

We have added to the main text that:

In general, the signal strength decreases with increasing chain length, as it is set by a combination of the degree of polarisation and decoherence (T_2 relative to C_{ij}) of all spins in the chain (See Supplementary Note 3). This limits the chain lengths that can be effectively used.

The quantitative equations for the signal strength for increasing chain length were given in Supplementary Note 3. We now added an explicit discussion of what limits the signal, and in particular how the T_2 coherence time becomes limiting far from the NV (due to instantaneous diffusion – see Supplementary Note 3).

3. For example, why are chains limited to 5? Is there some fidelity/polarization transfer infidelity? Can the authors quantify?

See previous comment. The 5-spin chain is an example. Longer chains can likely be formed, especially if one restricts oneself to spins that give strong signals. However, this is not required to map the 50-spin system in this work.

Note that longer chains that extend further from the NV become increasingly challenging due to the reduction of T_2 (see previous comment – and Suppl. Fig. 6).

4. A quick statement in the SI might be nice about why certain nuclear spins are polarized with PulsePol vs measurement/swap.

As detailed in Supplementary Note 1.2, initialisation by measurement or SWAP gate can yield higher initialisation fidelities, provided that high-fidelity selective control gates are available, but this is not effective for all spins.

5. In this work, the nuclear bath is polarized, freezing many interactions. Could similar schemes work with unpolarized baths? Understanding that coherences and flip-flops hurt.

In our experiment, the primary reason that flip-flop dynamics are frozen is due to disorder induced from the hyperfine interaction with the electron spin, which we generally keep in the $m_s=-1$ state.

In this scenario, the nuclear T_2 coherence (which plays a central role in the spin-chain sensing), does not strongly depend on the nuclear spin bath polarization that we create. A comparable statement is present in the main text (section 'experimental system', end of second paragraph).

For completeness, we note that the polarisation of the bath should indeed somewhat freeze or slow down the dynamics in addition to the hyperfine gradient of the NV (especially for spins that are practically uncoupled to the NV), but this is expected to be a secondary effect here and detecting it would require developing different experiments (like evolution of the system under the NV $m_s=0$ state for which the polarisation would be the leading factor for slowing flip-flops).

6. The discussion starting line 220 is a little unclear. Would consider rewriting.

We now explicitly state that the correlated spin frequencies and couplings “*are found to originate from the same branch of the network*” and make clearer that this leads to their unique identification (Section: spin-chain sensing, fourth paragraph).

7. I am curious how this scheme works when extending beyond just sensing the presence of the spins. Can similar sequences be used to utilize these distant spins for quantum applications? (perform gates?)

Yes, one of the prospects of this work is that the spin-chain sequence (possibly complemented by selective initialization of spins in the chain) can be used to read out spins after they participated in a quantum simulation. Two-qubit gates between nuclear spins can be realized via spin-echo sequences (see e.g. Randall et al., Science 2021), or can be mediated by the electron spin (see e.g. Abobeih et al. Nature 2022).

We added a sentence in the discussion section to further stress the potential for quantum gates and applications like quantum simulation.

8. Line 244 could use a reference for t_2^* (SI?)

We now include a reference to Bradley et al., (PRX 2019) at this sentence.

9. How does measurement of delta reduce ambiguity exactly? Line 248.

Each nuclear spin has a unique shift due its hyperfine interaction with the electron spin. We can use this as a label to identify a specific nuclear spin. In this way we know how many spins there are and which spins participate in specific measurements. A T2*-limited measurement (such as a conventional Ramsey or SEDOR spectroscopy such as in Fig. 2d) does not offer the resolution required to distinguish spins that are spectrally close. Our measurement of delta is more precise (T2-limited), alleviating this ambiguity and enabling a direct identification and labelling of individual spins by their specific delta value.

We adapted this sentence slightly to explicitly clarify the concept of 'labelling' the spins with delta.

10. Sentence starting line 250 unclear.

We changed 'frequency band' to 'frequency region' to enhance clarity.

11. There is a question on how certain the chains are reconstructed, and how uniquely spins are identified. What are the cases where this scheme can be confused/breaks down? Which spins can it not access?

First, regarding unique identification and potential confusion (mislabelling), our methods to improve resolution and correlate spin chains through the network, providing extra information for identifying spins and their connections. Very generally speaking, only measured chains that share a known starting (or ending) point (e.g. a spectrally isolated nuclear spin) can be merged with absolute certainty, as there is no ambiguity about this isolated part of the chain(s).

In practice, however, absolute certainty is not needed, and due to the disordered network, it should already be extremely unlikely for, e.g., two independent 4-spin chains that are identical (up to the Hz- resolution measurements of the four Delta and three C values) to exist. A theoretical or numerical analysis of the probability to misidentify a randomly drawn network (defined by the same some underlying properties), given a certain set of measurements, would be interesting (and non-trivial) and goes beyond this manuscript.

Note that in this particular case, we can verify the solution using knowledge of the interactions as function of spatial position $C_{ij}(r_i, r_j)$ (dipolar) and look for a corresponding spatial solution (see Fig. 5 and Suppl. Fig 3.) [18]. Because the problem is highly overdetermined (1225 interactions and $3 \times 50 - 4 = 146$ spatial coordinates), it is exceedingly unlikely that (1) two significantly different spatial configurations can produce the same measurement results, and (2) that a mislabelling (e.g. two spins are misinterpreted as a single one) can yield a valid spatial solution. This was numerically corroborated in reference [18], although precise statements about probabilities are difficult due to the complexity of the problem (a 146 parameter fitting problem!).

Second, regarding which spins cannot access, the advantage of the method breaks down if the nuclear spin that is detected has low T2 coherence time and therefore cannot be used as high-resolution 'sensor' itself (see also function CheckVertex (Sup. Section F)).

As nuclear T2 coherence times decrease further away from the NV center (instantaneous diffusion, see Suppl. Fig 6.), this limits the number of new spins that we can identify to be unique and thus add to our map of the network. We added a sentence to the methods section where the algorithm is described to link to this limiting behaviour more explicitly:

“Note that the measurement resolution, determined by the nuclear T_2 -time, is expected to decrease for spins further away from the NV center (See Supplementary Note 2). This eventually limits the number of unique spins that can be identified and added to the network map.”

12. Related to my other question. How far out (spatially) might one expect to extend the sensing? What about isotopically purified samples?

Supplementary table 1 shows the most likely spin positions, showing that for our current experiment the positions lie within a range of about 4 nm from the NV center. This distance is limited not by sensing sensitivity (the electron-nuclear coupling strength and the electron coherence), but rather by spectral crowding (Suppl. Fig. 6).

Regarding isotopic purification, the key general principle to be aware of is that the system formed by just an (point) electron spin and a surrounding C13 bath displays a form of scale invariance. All the spin-spin couplings (e.g. the frequency shift Δ and couplings C_{ij}) and all the line widths (i.e. $1/T_2^*$) scale linearly with the spatial isotope density (concentration). Therefore, the ratio between line separation and line width remains the same, and all the physics is unchanged with concentration (except for a change in absolute time and distance scales, which can be made arbitrarily large). In practice, however, at low enough concentrations other noise sources (other defects, magnets, etc) will dominate and limit the coherence. At high concentrations, one needs to consider the discreteness of the lattice and contact hyperfine due to the finite NV electron wavefunction. Additionally, calculating T2 echo values is non-trivial, as they depend on the system dynamics. Further analysis is beyond this manuscript.

We now explicitly explain the scale invariance for spectral crowding and the principles of our methods in the relevant section (Supplementary Note 2).

13. Why are only two spin chains measured? It seems some nuclei are missing? I recommend on Figure 5B to label/show the newly accessed spins compared to previous work.

Figure 5a shows a schematic example illustrating the concept of spin-chain merging (not a comprehensive overview of the measured data). We added *“Schematic illustrating the ...”* to the caption to clarify.

We now labelled the spins in figure 5b according to Supplementary Table 1 and added highlights to the newly mapped spins. We added to the caption:

“The spins are labelled according to Supplementary Table 1. Black circles indicate the 23 newly mapped spins compared to previous work [18].”

14. Why was DDRF not used?

As discussed in the experimental section, DDRF was used for the readout of some of the spins labelled 'direct' in Supplementary Table I (see also Supplementary Note 1.3). We now added "(DDRf)" in the first paragraph of the Experimental System section to clarify this.

Reviewer #3

Dear editors,

Thank you for this opportunity to review the manuscript "Mapping a 50-spin-qubit network through correlated sensing." The manuscript reports on double-resonance methods to measure the single-spin and two-spin coefficients of a (classical) Ising spin network. The authors experimentally demonstrate these methods by performing experiments on a nitrogen-vacancy center in diamond that is coupled to an ensemble of nearby nuclear spins. The experimental methods used are not per se novel, but the demonstration of sequential two-spin correlation blocks on chains of multiple spins and their deployment at scale in a real sample is technically impressive and non trivial. There is also novelty in how the data is processed (Sec. E). A another key achievement is extending their previous results (Nature 576, 411–415 (2019)) from 27 spins to 50 spins, offering opportunities for realizing quantum information processing protocols using solid-state spin qubits.

My main concern is that the manuscript and the figures are very difficult to read and to parse. I believe that it will be hard for scientists from other communities or those that are not immediately familiar with the previous work of the authors to understand the work and make use of these methods in other context. The manuscript includes many instances where greater specificity in language choice, proper definitions, and pedestrian explanations would help the reader better understand the work and its subtleties.

For these reasons, I suggest having the authors further revise ther text and figures to improve the clarity of their manuscript before making a final decision on publication. I foresee that a significantly edited manuscript would be suitable for publication in Nature Communications.

We thank the referee for their extensive feedback and suggestions to further improve the clarity of the manuscript. Below we address the suggestions point by point.

As a general note, we appreciate the detailed suggestions and believe that these have helped improve the clarity. At the same time, we hope that it is also appreciated that each reader might have different opinions about what explanations, examples, representations and visualizations they might find most intuitive (see e.g. the other reviewer comments), and that it is challenging to match all preferences simultaneously.

General comments:

1. Figures could be significantly edited to improve knowledge transfer to the reader (see comments below). Figures are generally too crowded (in both space and colors), hard to parse due to slight inconsistencies between main text, captions, and figures, and quite technical.

We updated the schematics for all the figures to improve consistency (see comments below for details).

Regarding the perceived technical and detailed nature of the figures, we believe that is appropriate for the type of work presented, which includes large complex pulse sequences, with many parameters, and large data sets. In particular, we think it is important that the figures contain enough detail for readers to reconstruct what the performed sequences are, as this provides a common ground for understanding and reproduction, regardless of the semantics of the descriptions.

2. Please revise your labeling and/or network representation conventions to insure consistency between your Hamiltonian and the network representation. I suggest independently labeling spectral classes and spin within each spectral classes, as well as avoiding the polychromatic representation of spectral classes. There are too many colors used for different purposes: color is meaningfully used to isolate a chain of spins within the network; however, Fig. 1 uses dark-purple for nodes, whereas Fig. 2 uses multicolor nodes. This coloring seems inconsistent/redundant as spectral classes are already labeled by the A_j coefficients.

To improve the connection between the Hamiltonian and the network visualisation, we now independently label the spins (the vertices in the network) and the spectral regions (coloured disks), which may contain multiple spins.

Additionally, we improved the consistency of the use of colour across the figures, to avoid unintended misinterpretation of the meaning of the colours. We kept the use of different colours in Figure 2, as they link the experimental frequencies in Fig. 2 d-g with the schematic in 2a-c.

3. Please revise the text so that all technical terms are properly defined, especially those that have ambiguous meanings, and that no jargon/loose terms are used. Whenever possible, add short explanations on what is implicitly assumed because it is common knowledge in your community, but might not be shared knowledge by all readers, e.g., describing in a few sentences how the SEDOR sequence gains information about the interaction strength between two spins, or how “the nuclear-spin state is mapped as a polarization difference on the NV electron spin, which is readout through spin-dependent fluorescent in a confocal microscope operating at 3.7 K (see Methods).” These short sentences help anchoring the reader on known facts without having them to seek information elsewhere.

We have carefully considered the description and textual definitions throughout the manuscript (see comments below for more details). We note that next to the textual descriptions, we provide the pulse sequences and equations for all relevant parts of the experiment, allowing readers to additionally apply their own intuition and semantics for translating pulse sequences to functionality.

To more explicitly describe how the SEDOR sequence provides information about the interaction strength, we added the sentence:

“The signal is acquired by mapping the resulting nuclear spin polarisation, for example at frequency ν_{A_i} , on the NV electron spin and reading it out optically (Supp. Note 3).”

to the manuscript, including a reference to the existing Supp. Note 3, which gives a stepwise mathematical derivation that supplements the provided pulse sequence (Fig. 2b) and textual explanation.

Abstract

4. “of current methods”: what methods are you referring to?

As explained in more detail in the main-text introduction, this refers to the methods as described in references 18 and 19 (and previous work referenced therein). Particularly relevant is the method of reference 18, which we describe in Fig. 1a and 2b, and the discussion of those figures. We note that Nature Communications has unreferenced abstracts, so that we provide the detailed and referenced description of the state of the art in the introduction, rather than in the abstract.

5. “complexity of these spin networks”: “complexity of a network” is ambiguous, please revise. I assume that by “size” you mean the number of spins/nodes, and by “complexity” the number of edges. Also, how is “complexity” constrained by spatial resolution?

We deleted the superfluous word ‘complexity’, as in this context ‘size’ and ‘complexity’ imply the same thing (the number of connected spins).

6. “through high-resolution correlated sensing schemes”: the usage of “correlated sensing” is ambiguous, as it is not clear what is correlated with what (see below for further discussion); consider using a different term or being more specific.

We added extra context regarding the use of the word ‘correlated’ to the discussion in the main text (Section: spin-network mapping – see comments below). A detailed explanation in the abstract is not possible due to the word count.

7. “with high spectral resolution”: the usage of “high” is ambiguous and relative; rather emphasize that your methods are improving the spectral resolution over previously demonstrated methods; also specify what increase in size your methods enable, e.g., going from 27 spins to 50 spins.

The precise quantitative improvements and comparisons to the state of the art are given in the main text (introduction and in the results, e.g. Fig. 3).

We chose to not include such details in the abstract. Instead, we use a layered structure, in which the introduction provides such details. We consider this a subjective style choice that does not impact the contents of the paper. Note that the word limits make the abstract unsuitable for much more detail.

8. “opportunities for quantum simulation”: be more specific to make the statement more convincing as accessing more spins without proper control and interactions does not necessarily lead to greater simulation capability, and classical spin Hamiltonian (1) does not allow for universal quantum simulation, e.g., “opportunities for simulating spin dynamics in Ising spin networks” (same comments for references about quantum simulations in the introduction).

The spin Hamiltonian (1) is the Hamiltonian of our system for the specific case of free evolution with the electron spin fixed in the $m_s=-1$ state. This Hamiltonian includes the interactions C_{ij} and the frequencies A_i that we aim to characterize to map out the network.

However, this Hamiltonian does not include the control fields (of form I_x and I_y) for the nuclear spins and the electron spin. Nor does it include the full hyperfine tensor and the flip-flop interactions between the nuclear spins, which both provide additional control tools by manipulating the electron spin state (see equation 1 in Supplementary Note 1 (‘Hamiltonian’) for the more general description). Together with the control fields, the spin Hamiltonian provides a universal quantum gate set for simulation and other applications (albeit with non-ideal gate fidelities, of course). This is detailed for example in Refs [1], [16] and [32].

To clarify how this system can allow for (future) quantum simulation we have added a statement to the conclusion:

“Combined with control fields [1,16,32], the methods developed here provide a basis for universal quantum control and readout of the network, which has potential applications in quantum simulations of many-body physics [1].”

Introduction

9. “a key open challenge is spectral crowding”: given that the abstract states that the problem is “spectral resolution of current methods”, the root cause of “spectral crowding” should be better explained. My point is that spectral crowding seems to depend on both the “spectral resolution of current methods” being limited AND the distribution of spectral contributions in the sample. Because the number of spins with the same frequency increases with the distance away from the central spin (the volume of a spherical shell approximately scales as r^2), spectral crowding seems to be more likely for far-away/low-frequency spins. Thus, spectral crowding depends on both resolution and bandwidth of detection method (smallest frequency that can be accessed), which in turn depends on the density of the sample.

In this context, we define spectral crowding as two or more spins having a precession frequency difference that is small compared to their linewidths (set by the inverse of T_2^*). The equation is given in main text line 209 and detailed in Supplementary Note 2. This is a property of the sample spins (their frequency distribution and linewidth). Experiments like in Refs 18 and 19 are limited by this overlap and do not unambiguously determine how many spins are present and which ones contribute to signals observed at a given frequency f . Our method uses correlated measurements (chains) to obtain the desired information despite spectral crowding, as well as manipulation of the sample (double echoes) to enhance the spectral resolution by improving the sample coherence (this is a property of the sample in combination with how it is manipulated, i.e. the method).

If this statement is true, because “spectral crowding” depends on sample properties, to access large system size for “quantum simulation”, you might then choose to either increase the density of nuclear-spin defects in the sample (because nuclear spins close to the NV are less likely to spectrally overlap) or access nuclear-spin defects at greater distance. It is not clear if your method performs equally well for the two cases. If your approach depends on the density of nuclear spins in the sample, then the importance of your results would be less about the total number of spins (50 spins) than the total number of spins in a given detection volume (the density). In short, some explanation is lacking for the reader to understand the relationships between spectral resolution, spectral crowding, and network size (some of it is implied, but not explicitly addressed).

The reviewer asks to clarify what happens if the nuclear spin concentration (density) would be altered, and suggests to use a higher C13 concentration because “nuclear spins close to the NV are less likely to spectrally overlap”. It is important to understand that the system formed by a (point) electron spin and a surrounding nuclear spin bath displays a form of scale invariance. All the spin-spin interactions and all the line widths scale linearly with the concentration. Therefore, the ratio between line separation and line width remains the same. Spectral crowding and all the physics are unchanged with concentration (except for absolute time and length scales). In this case, the number of spins that can be mapped is independent of concentration.

Note that, in practice, this invariance breaks down at sufficiently high concentrations, as one needs to consider the discreteness of the lattice, the extra contact hyperfine due to the finite NV electron wavefunction, and the practical limitations to the MW power. Additionally, calculating T2 values is non-trivial, as they depend on the system dynamics. Therefore, the question what the optimal concentration would be for a given goal (like mapping the largest number of spins), is not trivial and beyond this manuscript.

To address this comment (see also comment 12 from reviewer 2), we now explicitly mention the scale-invariant nature of this problem, and the dependence on concentration in Suppl. Note 2 (which already included the relevant equations).

For completeness, we note that the reviewer is right that using different type of samples/devices with less spectral crowding, could lead to mapping and control of more spins. Examples are to use a material like silicon carbide, which has two spin species that naturally are separated in frequency, or to apply a strong field gradient. However, (experimentally or theoretically) studying the plethora of exciting possibilities to go beyond the 50-spin benchmark set here is beyond the current manuscript.

10. as mentioned above, “network connectivity” is ambiguous, as it might refer to whether a vertex is connected with another one, which is always true for all-to-all coupled spin networks. Please be more specific, e.g., “that measure the frequencies and couplings of a network of spins”.

We now added a statement defining the connectivity and the concept of resolvable couplings in the context of our work (Section spin-network mapping, second paragraph):

"Although in principle all spins are coupled to all spins, we draw edges only for strong, resolvable, spin-spin couplings, defined by: $C_{ij} > 1/T_2$, where T_2 is the nuclear Hahn-echo coherence time (~ 0.5 s) [16]. The network connectivity constitutes the presence (or absence) of such resolvable couplings."

11. "measure chains of coupled spins": it is not clear what you mean by "measuring chains". Please be more specific when describing the key concept of your method. Also, "these spin chains remove ambiguity" is ambiguous, as "spin chains" is not the proper subject of the verb "remove". (Generally, look for similar instances of loose/ambiguous language).

We added a statement explicitly defining when spins are considered to be coupled (resolvable coupling, see previous comment). We also adapted the sentence, so that the 'mapping of spin chains' becomes the proper subject of the sentence.

Section A:

12. The sum Eq. (1) runs up to $N/2$, which is ambiguous for N odd. Consider using $\sum_i^N \sum_{j=\{i+1\}}^N$.

We have adapted the main text and supplementary info to fix this issue.

13. Eq. (1) is a priori valid for any Ising spin network, including both electron and nuclear spin defects. Specify whether results presenting in the manuscript are generally valid for all spins or only applicable to nuclear spins (even though the demonstration is made for nuclear spins).

While the actual experiments (starting at 'experimental system') are performed on a particular system, the spin chain methods are expected to be applicable in a large class of systems (potentially with adaptations), including other spin defects, and both electron and nuclear spins systems.

To clarify this, we have adapted the related sentence in the conclusion to:

*The methods can be applied to a variety of systems in different platforms, **including electron-electron spins networks** [7–15, **43**].*

Where bold face marks the changes. A discussion about all other possible types of spin systems, and what methods would be most effective in those hypothetical cases, is beyond this manuscript. Reference 43 is a recent preprint that was mentioned in an *author's note* in the previous manuscript version, but such notes are not allowed in the style of Nature Comm., so that we now include it here.

For completeness we note that the electron-nuclear sequence (Fig. 3) is specific to systems in which the origin of the spectral shifts (here the NV electron spin) can be inverted or toggled, so that it can be recoupled.

14. Under which conditions is Eq. (1) valid?

As explained in the second paragraph of the 'experimental system' section and Supplementary Note 1.1, Eq. 1 is an accurate description of the free-evolution Hamiltonian of our spin network at high B-field and for the electron spin in the $m_s = \pm 1$ state.

15. "In general, the number of frequency vertices is smaller than the number of spins, as multiple spins might occupy the same frequency vertex.": A distinction would ideally be made between the true underlying physical all-to-all spin network representing Hamiltonian (1) (for which all coefficients are non-zero) and the estimated network reconstructed from measurements with finite resolution (where some coefficients might be indistinguishable from zero or overlapping). For example, one node in Fig. 1 is labeled as A_23 which is very confusing, as it does not match definitions in Eq. (1).

As explained at comment #3, we have now adapted the nomenclature and schematics to make this distinction explicit.

Paragraph 84-92:

16. "State-of-the-art" demonstrations: it is quite interesting that the authors refer to their previous work as "state-of-the-art" in plural form. It be more appropriate to use "In our previous work" or "In our more recent work".

We changed the sentence to singular instead of plural.

17. Please add references to further expand on existing approaches to spin network matpping, as well as prior uses of SEDOR-like sequences, possibly including prior work with NV centers for both electron and nuclear spins, as well as prior work in NMR.

We added three additional references to the introduction to provide more background regarding previous approaches:

- Zopes et al., Nat. Commun. 2018
- Zopes et al., PRL 2018
- Shi et al., Nat. Physics 2014

We are not aware of any traditional NMR work with the capability to map spin networks of individual spins in this way. Covering the vast literature on traditional NMR sequences and methods to e.g. protein structure determination is beyond the scope of this work.

18. Please provide enough information for the reader to understand your previous work without having to access and read your previous paper or Methods section. E.g., in this paragraph reviewing previous work and describing its limitations, it is not clear how the NV center is involved and whether these "SEDOR" sequences are "correlated sequences" in the way that was defined in the abstract. It is also not clear if by " T_2, T_2^* ", you are referring to the decoherence time of the NV electron spin or of the nuclear spins (5 ms is mentioned in Sec. D)? If the decoherence time of the nuclear spins, does it change among spins, what limits its

value, what is its (typical) value in your current system? What is its associated spectral resolution?

We added the word 'nuclear' to specify that the T_2 and T_2^* times given are for the nuclear spins. We also added a typical value of the nuclear T_2 time (which is relevant for this section) with a reference.

To avoid confusion, we now include a sentence on the role of the NV center in the SEDOR measurement already at this stage in the manuscript. Note that, in principle, discussion at this point is agnostic to how this is done exactly, and the actual implementation details follow in the section 'experimental system'.

SEDOR measurements also correlate frequencies and couplings together, and thus the word correlated can be used (this is semantics). However, they only correlate in a pairwise fashion, which provides less information than our methods based on spin chains. As usual, the spectral resolution is inversely proportional to the coherence time.

19. Consider adding a clear upfront statement about what was done in your previous work and what is new with this work.

After careful considering this, we conclude that the advance beyond our previous work, and that of others, is already stated multiple times, with increasing levels of extra details, in the abstract, introduction, main text and Supplementary notes.

20. Line 133: "Correlated": What do you mean by "correlated"? "Correlated sensing" is mentioned multiple times in passing without a clear definition. I understand that the recoupled spin-echo sequence correlates the spin states of two spins by mapping the operator $I_z * I_d$ to $I_z * I_j$ with a modulation factor that depends on the interaction strength; however, "correlated" might be understood differently by different readers, cf. correlation spectroscopy of Laraoui2013. Please make sure that jargon is properly defined and its usage justified.

Our measurements are correlated in the sense that they provide information about what (sets of) spins are coupled to each other. For example, performing the spin-chain sequence on 3 frequencies, returns the correlated list {A1, C12, A2, C23, A3} directly identifying that a single spin with frequency A2 is coupled to both spin 1 and spin 3. Conversely, and experiment with two independent (uncorrelated) sequences would not provide such information, as it does not distinguish between a connected three-spin chain or two independent (uncoupled) two-spin chains. We adapted the relevant paragraphs to clarify this more explicitly.

We note that this use of the word "correlated" is closely analogous to the standard use in NMR, see for example COSY (CORrelation Spectroscopy), even though the sequences, type of system and situations considered here are very different. In particular, the observation of a correlation between multiple frequencies (in SEDOR or COSY) indicates that these frequencies originate from the same (group of) spins, yielding additional information beyond the (one-dimensional) spectrum. Clearly, the general words "correlation" and "correlation spectroscopy" are found in various contexts where a relationship between systems, measurement values or variables is probed, but it seems out of scope to discuss other uses in this work.

21. Line 145: “This allows for the merging of chains”: What do you mean by “the merging of chains.” How is this done in practice?

With the ‘merging’ of chains we mean the process of identifying if two independently measured chains share a common overlapping section. In this way one can reconstruct the (branched) structure of the network by measuring multiple chains and merging them. The protocol on how to do this is outlined in the Methods section (‘Network reconstruction’). We now added an explicit reference (to the Methods section) in the last paragraph of the ‘spin-network mapping’ section. We also made the process more explicit in the schematic of Fig. 5A.

22. Section E: Does the main text has an explicit reference to Fig. 4e? Results reported in Fig. 4d-f seems very interesting, but it is hard to understand them as the main text offers too little explanation.

We added a reference in the main text to figure 4e. We added a sentence explaining the workings of the 2D experiment in more detail in the last paragraph of the ‘high-resolution measurement of spin frequencies’ section:

“After the t_1 evolution, the hyperfine shifts Δ_i are encoded in the z-expectation value of each spin, effectively modulating the nuclear-nuclear couplings observed in t_2 .”

Figure 1

23. This figure would benefit from further editing: The fully-connected network is confusing because many nodes have the same labels (e.g., A_1). The (a-b-c-d) labels are also hidden in the network and not aligned. I suggest having 4 disjoint sub-figures with the (a-b-c-d) labels in black at the upper left corner (following typical APS/Nature publication standards) presenting different networks or emphasizing different features of the same network.

We implemented this suggestion, splitting up the network in four more traditional subfigures labelled a,b,c and d. This does not change the actual information, but we hope it improves accessibility.

24. The vertex represented as purple disks represent “spin precession frequencies”, but the edges connect spins.

The distinction between frequencies and spins with those frequencies has been made more explicitly (see comment 2).

25. There are too many colors that are hard to distinguish from one another.

We adapted the schematics so that they are consistent across figures. Colours are used to highlight different spectral regions.

26. “Mapping complex spin networks.” I do not understand the usage of complex, as what is shown is a standard spin network.

The word ‘complex’ was intended to indicate the difference with the less complex networks in state-of-the-art experiments, both in terms of size and spectral crowding. As it is not critical, we removed the word ‘complex’ to avoid confusion.

27. “one spin in each vertex”: this nomenclature is confusing, as it does not directly connects with the Hamiltonian from Eq. (1). An easy fix would be to revise Eq. (1) to include subindex, e.g., $A_{\{jk\}}$ labeling spectral classes, and spins within each spectral classes; however, this choice might be confusing when considering the $C_{\{jk\}}$ which are specified for spins rather than spectral classes.

We addressed this ambiguity by referring to the coloured disks as ‘spectral regions’, which can be occupied by a spin if its precession frequency A_i lies within that spectral region (see also comment 2).

28. In Fig. 1(b), you might have the two A_2 spins labeled as $A_{\{2,1\}}$ and $A_{\{2,2\}}$ upon lifting the degeneracy. Also, if the blue spin was connected to the purple spin then $C_{\{2,4\}}$ would be defined twice, which is very confusing.

See previous comment.

29. “By measuring chains”: this language is once again ambiguous

Given the detailed descriptions throughout the rest of the manuscript (see e.g. Fig. 2), and the adaptations made based on the other comments, we consider that “measuring chains” is sufficiently unambiguous.

30. The graph shows $C_{\{1,4\}}$ with an X, but the caption says “the coupling $C_{\{34\}}$ is not directly accessible from the spin $A_{\{1\}}$.” Why is XX used first, then XY? What if XX was used instead of XX?

We adapted the caption to mention that all couplings belonging to Spin 4 are hard to access, by adapting the statement to:

“...couplings belonging to Spin 4 are not directly accessible...”

This avoids confusion about the relation between C34 and C14.

Figure 2

31. this figure is very crowded and the different colors are overwhelming. Please simplify.

We made small modifications in the visualisation and improved the clarity and the consistency of colour use (see comment 1). As stated at comment 1, we feel that further simplification would cause the loss of important information.

32. “coloured spheres” -> “coloured disks”

We updated this.

33. to be consistent with Fig. 1, A_j labels should be associated with spectral classes rather than spins

See comment 2, where this is addressed.

34. Because there is only spin per spectral class, it might be sufficient to have labeled nodes rather than dots within colored nodes. I suggest removing the C_{jk} from circles and having them labeling the edges.

For consistency across figures, we prefer to keep the more general visualisation (even if this particular figure shows only one spin per spectral class).

35. (b) What does “RO” refer to? How is “ t ” in “XX” and “RO” chosen? Could it be that t_{12} refers to the time of the correlation block applied to spin 1-2? Is “ t ” fixed or varying? A variable might be denoted by \vec{t} . Why is there an arrow pointing up?

We added in the caption that “RO stands for readout” plus a reference to the methods section, where the “RO” block is explained.

The time “ t ” can be varied. See e.g. Fig. 2d-g for examples where “ t ” is varied. We added a sentence explaining how the free evolution time “ t ” is chosen for the measurement in 2d. We changed t_{12} to italic and added ij-subscripts in the grey detail blocks for an unambiguous definition. The arrow conceptually illustrates the direction in which the spin expectation values are mapped to the next spin in the chain.

36. colors of circles make it hard to read the labels.

We have adapted the graphic style to make labels more readable.

37. “correlating 5 spin frequencies and 4 spin-spin interactions”: the sequence correlates the spin states of 5 spins; how does it correlate frequencies and interactions? Please be specific in your language to avoid introducing confusion. “To probe the connection at “A4”? What does that mean? (d-g) It would be beneficial for the reader to describe how the sequence is deployed to extract the relevant coefficients, e.g. “We sweep the frequency of the recoupling pulse on the second spin to extract its resonance frequency. We then sweep the time of the recoupling sequence to extract the coupling strength between the first and second spin. We repeat these steps for all pairs of spins in the chains.”

The use of the word “correlation” was discussed in comment 20. Probing the connections is done by sweeping both the RF frequency and free evolution time. We adapted the figure caption (2d-g) to better convey the goal of extracting couplings and frequencies by adapting the sentence:

“Experimental data, sweeping the frequency RF_N of the recoupling pulse (left) to detect the frequencies of spins coupled to Spin 1, and varying the free evolution time $t_{N,N-1}$ (right) to extract their coupling strengths (for $N = 2,3,4,5$)”

Figure 3

38. What is “RO”?

We now explicitly state that it is an abbreviation for “readout”. See comment 35.

39. Why is there a need to represent the hyperfine interaction by the blue dotted line? To be consistent with the main text, do you mean the “nuclear-spin frequencies ω_i ”?

The line indicates the interaction. We now explicitly label the line with the frequency shift due to the interaction Δ_i to avoid confusion.

40. “Between the nuclear spin and the electron” -> “Between the nuclear spin and the electron spin.”

We implemented this suggestion.

41. “Picking up signal from the interaction” is too vague to be meaningful. The diagram would ideally indicate a frequency/detuning to connect with (c). Is the electron or nuclear-spin frequency being swept?

We changed “signal” to “a phase” to be more precise. In panel c the time t is swept as indicated on the x-axis (the frequencies of the pulses for the electron and nuclear spins are not swept).

42. What is the coherence time of 0.36(2) s limited by? Is that the T_2^* of the nuclear spin?

We now explicitly label this time as T_2 , since it results from a spin-echo type experiment. The limit to this coherence time is non-trivial, and includes a second-order effect due to the hyperfine interaction, as is discussed in detail in the main text (section high-resolution measurement of spin frequencies, fourth paragraph) and further detailed in Supplementary Note 4.

43. that is ~ 75 times improved”: with respect to what reference?

We added: “with respect to (c)” to the figure caption.

Figure 4

44. How is a spin cluster defined?

We changed the figure title to: “Two-dimensional spectroscopy of spectrally crowded spins” to reflect the content better.

45. Revise A_{23} to be consistent with Eq. (1)

We adapted all figures for consistency with Eq. (1) (see above comments).

46. “chained readout”: it is a nice term, but ambiguous: chained readout of what?

We changed “which is used for chained readout” with “which is used to transfer the signal to the electron spin for readout”.

47. “of the frequency region A_{23} ”? Is A_{23} a frequency or a frequency region?

We updated the definition of the frequency region to: A_2 , A_3 , and stated that these frequencies are approximately equal ($A_2 \approx A_3$).

48. “reveals the correlation between Δ_i and spin-spin couplings”: the main text makes a similar statement in Line 283, but it is done too fast without a clear explanation of why Δ_i is correlated with spin-spin couplings.

In the first evolution time (t_1) the spins gain phase at a rate given by Δ_i . This effectively modulates the z-expectation value of the spins, before the second free evolution time (t_2) starts. Hence, the nuclear-nuclear coupling is modulated by (i.e. correlated with) its value of Δ_i (with T2-limited resolution). We added amore explicit explanation:

After the t_1 - evolution, the hyperfine shifts Δ_i are imprinted in the z-expectation value of each spin, effectively modulating the nuclear-nuclear couplings observed in t_2 .

49. More explanation is needed to understand Fig. 4e.

We now explicitly refer to Fig. 4e at the point in the text where it is described. Also, we have split-up the (joint) pulse diagram into two separate pulse diagrams, so that it is more clear what experiment is performed in Fig. 4d and 4e.